# Spatial and temporal variation of $\delta^{13}$C values of methane emitted from a hemiboreal mire: Methanogenesis, methanotrophy, and hysteresis

Janne Rinne[1,2], Patryk Łakomiec[1], Patrik Vestin[1], Joel D. White[1], Per Weslien[3] Julia Kelly[4], Natascha Kljun[4], Lena Ström[1], Leif Klemedtsson[3]

[1]Lund University, Department of Physical Geography and Ecosystem Science, Lund, Sweden
[2]Natural Resources Institute Finland, Production Systems Unit, Helsinki, Finland
[3]University of Gothenburg, Department of Earth Sciences, Gothenburg, Sweden
[4]Lund University, Centre for Environmental and Climate Science, Lund, Sweden

*Correspondence to*: Janne Rinne (janne.rinne@luke.fi)

**Abstract.** The reasons for spatial and temporal variation of methane emission from mire ecosystems are not fully understood. Stable isotope signatures of the emitted methane can offer clues to the causes of these variations. We measured the methane emission ($F_{CH4}$) and $^{13}$C-signature ($\delta^{13}$C) of emitted methane by automated chambers at a hemiboreal mire for two growing seasons. In addition, we used ambient methane mixing ratios and $\delta^{13}$C to calculate a mire-scale $^{13}$C signature using a nocturnal boundary-layer accumulation approach. Microbial methanogenic and methanotrophic communities were determined by a captured metagenomics analysis. The chamber measurements showed large and systematic spatial variations in $\delta^{13}$C-CH$_4$ of up to 15 ‰ but smaller and less systematic temporal variation. According to the spatial $\delta^{13}$C-$F_{CH4}$ relations, methanotrophy was unlikely to be the dominating cause for the spatial variation. Instead, these was indication for the substrate availability of methanogenesis to be a major factor explaining the spatial variation. Genetic analysis indicated that methanogenic communities at all sample locations were able to utilize both hydrogenotrophic and acetoclastic pathways and could thus adapt to changes in the available substrate. The temporal variation of $F_{CH4}$ and $\delta^{13}$C over the growing seasons showed hysteresis-like behavior at high-emission locations, indicative of time-lagged responses to temperature and substrate availability. The up-scaled chamber measurements and nocturnal boundary-layer accumulation measurements showed similar average $\delta^{13}$C values of -81.3 ‰ and -79.3 ‰, respectively, indicative of hydrogenotrophic methanogenesis at the mire. The close correspondence of the $\delta^{13}$C values obtained by the two methods lend confidence to the obtained mire scale isotopic signature. This and other recently published data on $\delta^{13}$C values of CH$_4$ emitted from northern mires are considerably lower than the values used in atmospheric inversion studies on methane sources, suggesting a need for revision of the model input.

## 1 Introduction

Methane ($CH_4$) is the one of the three main drivers of anthropogenic climate change. Its sources include both biological and anthropogenic processes, with the most significant natural source being wetland ecosystems (Ciais et al., 2013). As changing climate may influence global $CH_4$ emission from wetlands, a mechanistic understanding of the processes behind these emissions is crucial.

The $CH_4$ emission rates from wetlands are controlled by $CH_4$ production (methanogenesis), $CH_4$ oxidation (methanotrophy), and the transport of $CH_4$ from peat into the atmosphere (e.g. Lai, 2009). A fundamental factor for $CH_4$ production by Archaea is the availability of substrates, as $H_2$ or acetate for hydrogenotrophic or acetoclastic methanogenesis, respectively (e.g. Lai, 2009). Furthermore, temperature is a key driver of the $CH_4$ emission rate via its effect on microbial activity, as seen by the incubations of peat samples conducted at different temperatures (Juottonen at al., 2008). Water table position and the presence of alternative electron acceptors can also influence the spatial or temporal behavior of $CH_4$ production (e.g. Serrano-Silva et al., 2014). A part of the produced $CH_4$ is commonly oxidized in the wetland, and thus not emitted into the atmosphere (e.g. Larmola et al., 2010). This methanotrophy is caused by methanotrophic micro-organisms (bacteria), and it may also be dependent on temperature (Serrano-Silva et a., 2014). Finally, $CH_4$ can be transported from the anoxic layers to the atmosphere by three different mechanisms: diffusion through the peat matrix, ebullition, and plant mediated transport (Lai, 2009). The later can be further divided into passive diffusive transport and active convective transport (Brix et al., 1992).

The observed $CH_4$ emissions from wetland ecosystems exhibit both temporal and spatial variations, which reflect the variation in the above-mentioned processes, often in tandem. Typically, $CH_4$ emission rates vary spatially over short distances following surface microtopography (e.g. Riutta et al., 2007; Keane et al., 2021), and related differences in vegetation characteristics. The highest emission rates are commonly observed in wetter locations, with abundant aerenchymatous vegetation, whereas the lowest emission rates are observed at dry hummocks or inundated locations (e.g. Riutta et al., 2007, Keane et al., 2021). This microtopography-scale spatial variation in $CH_4$ emission can be caused by differences in the methanogenesis, methanotrophy, or transport pathways in these different locations (Joabsson et al., 1999; Joabsson & Christensen 2001).

Temporally, we commonly see a seasonal cycle in the $CH_4$ emission rates, with the highest emission rates in late summer (Rinne et al., 2018; Heiskanen et al., 2021b; Łakomiec et al., 2021). This seasonal variation has been associated with the seasonal cycle of peat temperature, substrate availability, and transport pathways (Rinne et al., 2018; Chang et al., 2020; 2021). Diel variation of $CH_4$ emission rates has also been observed in wetlands with vegetation such as *Phragmites*, *Typha*, and *Nymphaea* that exhibits pressurized airflow into the root systems, (Kim et al., 1998; Kowalska et al., 2013), whereas

wetlands with vegetation that exhibits diffusive air transport show little or no diel cycle in their $CH_4$ emission (Rinne et al., 2007; Jackowicz-Korczyński et al., 2010; Kowalska et al., 2013). In many cases the predominance of any one cause for temporal variation in $CH_4$ emission may be difficult to verify, as the variation of these different processes may lead to similar variations in the resulting $CH_4$ emission rate (Chang et al., 2021).

$CH_4$ emitted from different sources (e.g. wetlands with different methanogenic pathways, waste, ruminants, termites etc.) is characterized by different isotopic composition (Miller, 2005; Hornibrook 2009), and this isotopic composition can offer clues to the processes behind these emissions. The major component of $CH_4$, carbon, has two stable isotopes, $^{12}C$ and $^{13}C$, which make up 98.9% and 1.1% of carbon in nature, respectively. While different isotopes of the same element behave chemically identically, their different masses cause differences in their diffusion rates, and in the rates of many chemical and biological processes. This will lead to differences in the isotopic ratios of $CH_4$ as it goes through methanotrophy, methanogenesis or transport from the anoxic peat layers to the atmosphere

In mire ecosystems, which are defined as vegetated wetlands with capability for peat formation (Lindsay, 2018), the $^{13}C$ signature, or $\delta^{13}C$ value, of emitted $CH_4$ depends on its production pathway, and subsequent transport and oxidation (Hornibrook, 2009). Of the two dominating methanogenic pathways in wetlands, hydrogenotrophic methanogenesis typically produces $CH_4$ that has lower $\delta^{13}C$ value than $CH_4$ produced by acetoclastic pathway (Hornibrook, 2009). The first mentioned typically produces $CH_4$ with $\delta^{13}C$ value in the range from -110 ‰ to -60 ‰ and the latter one from -60 ‰ to -50 ‰ (Whiticar, 1999; McCalley et al., 2014). Furthermore, microbial oxidation of $CH_4$ can shift the emitted $CH_4$ to have higher $\delta^{13}C$ value, as microbial methanotrophy prefers $^{12}C$-$CH_4$ (Hornibrook 2009). Thus, the $\delta^{13}C$ values of the emitted $CH_4$ can be used as an additional constraint when interpreting the observed $CH_4$ emission rates to disentangle the processes responsible for the spatial and temporal variation in $CH_4$ emission. For example, recent analysis has shown hysteresis-like behavior between surface temperatures and $CH_4$ emission rates in mire ecosystems, and the possible causes of this phenomenon are debated (Chang et al., 2020; 2021; Łakomiec et al., 2021). Similar hysteresis-like behavior has also been observed between photosynthesis and $CH_4$ emission rates (Rinne et al., 2018). Stable isotope signatures of emitted methane can constrain our hypotheses on the causes of these behavior by refutation or corroboration.

In this study, we analyze the observed spatial and temporal variation of $CH_4$ emission rates from a hemiboreal mire ecosystem and its $\delta^{13}C$ values to understand the causes of these variations. We aim to shed light on the relative importance of methanogenesis and methanotrophy for the spatial variation in the $CH_4$ emission rate, and the roles of precursor substrate availability and temperature for the seasonal variation of the $CH_4$ emission rate. We also use taxonomy data to characterize the methanogenic and methanotrophic microbial communities in the mire to reveal the potential of methane production via different pathways as well as microbial methane oxidation.

In order to interpret the variation in CH$_4$ emission rates and their $\delta^{13}$C values, we have formulated a conceptual framework with different simplified hypotheses for the causes of the spatial and temporal variations of methane emission rates. From these we have deduced expected relations between CH$_4$ emission rates and their $\delta^{13}$C values, that are used to guide the data analysis and interpretation.

## 2 Conceptual framework

We will consider two commonly observed phenomena in the variation of CH$_4$ emission rates from mires. First, there is a spatial variation at the microtopographic level, with the lowest emissions from dry hummocks and inundated ponds and highest emissions from wet lawns (e.g. Riutta et al., 2007; Keane et al., 2021). Second, there is a temporal variation at the seasonal scale, which lags the cycle of air and peat surface temperature and GPP, but follows the temperature of deeper peat (e.g. Rinne et al., 2018; Chang et al., 2020; 2021; Łakomiec et al., 2021).

We can have two simplified hypotheses regarding the processes leading to the small-scale spatial variability of the CH$_4$ emission rate. In the first hypothesis on spatial variability (HS1), we assume that the production of CH$_4$ beneath wetter and drier surfaces is equal but that oxidation by methanotrophic organisms in the oxic layers leads to lower emission of CH$_4$ from the drier surfaces compared to the wetter surfaces (Figure 1). In the second spatial hypotheses (HS2), we assume that the differences in CH$_4$ emission rate between wet and dry surfaces reflect differences in CH$_4$ production due to differences in the substrate availability for methanogenesis. While both hypotheses lead to similar differences in the CH$_4$ emission rates between the wetter and drier surfaces, their relation to the $\delta^{13}$C values are different. HS1 would lead to negative correlation between CH$_4$ emission rate and $\delta^{13}$C value of emitted CH$_4$, because enzymatic reactions associated with methanotroph metabolism consumes preferentially $^{12}$CH$_4$, resulting $^{13}$C enrichment of residual CH$_4$. HS2, on the other hand, would lead to positive correlation between CH$_4$ emission rate and its $\delta^{13}$C value, because CH$_4$ production in conditions with better substrate availability, typically associated with higher methane emission rates of more productive mires, leads to CH$_4$ with higher $\delta^{13}$C value than in lower substrate availability (Chanton et al., 2005). The better substrate availability can be associated with acetate availability for acetoclastic methanogenesis or better energetics for hydrogenotrophic methanogenesis (Penning et al., 2005; Hornibrook 2009). Thus, the two hypotheses lead to distinctly different predictions on the relationship between CH$_4$ emission rate and its $\delta^{13}$C value (Hornibrook 2009). As a zero hypothesis (HS0) we may have e.g. a mixture of the above mentioned processes contributing to the spatial variability of CH$_4$ emission. In this case we may observe no systematic co-variation between CH$_4$ emission rate and $\delta^{13}$C values.

For the seasonal variation of CH$_4$ emission rate, we can hypothesize that the variation is either due to the seasonal development of temperature, or that it is modified heavily by the availability of substrates for methanogenesis (Chang et al., 2020; 2021). In the first hypothesis on the temporal variation (HT1), we assume that the temporal variation is due to the seasonal change in peat temperature. As this does not change the δ$^{13}$C value of emitted CH$_4$, there will be no temporal correlation between CH$_4$ emission rate and its δ$^{13}$C value (Figure 2). In the second temporal hypothesis (HT2) we assume that the seasonal cycle of the CH$_4$ emission rate is due to the changes in substrate availability. This may be via changes in availability of H$_2$ for hydrogenotrophic methanogenesis or by availability of acetate for acetoclastic methanogenesis. Thus, the changes of substrate availability may or may not include changes in the methanogenetic pathway. The HT2 would lead to positive correlation between CH$_4$ emission rate and its δ$^{13}$C value. In the third temporal hypothesis (HT3) we assume that there are significant time lags between the seasonal cycles of the drivers of CH$_4$ emission rate, i.e. temperature and substrate availability, which leads to hysteresis-like behavior in the relationship between CH$_4$ emission rate and its δ$^{13}$C value.

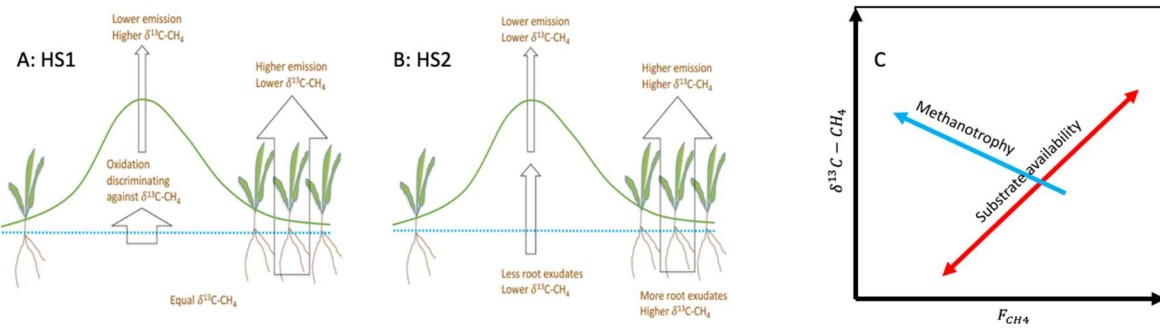

Figure 1: Spatial variation of methane emission based on two hypotheses: A: HS1, variation is due to methanotrophy; and B: HS2, variation is due to methanogenesis and substrate status. Resulting relations between δ$^{13}$C-CH$_4$ and F$_{CH4}$ are shown in panel C.

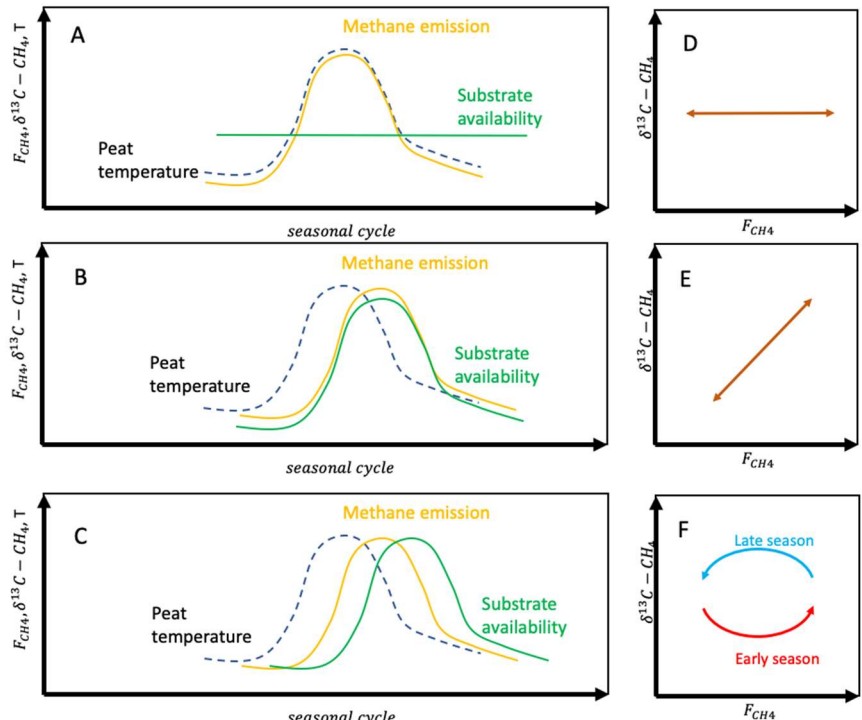

*Figure 2: Seasonal variation of methane emission with hypotheses on controlling processes (A: HT1; B: HT2; C: HT3) and resulting relations between $\delta^{13}C$-$CH_4$ and $F_{CH4}$ (D-F).*

## 3 Methods

**3.1 Study site and ancillary measurements**

We conducted the measurements at Mycklemossen mire (58°21'N 12°10'E, 80 m a.s.l., Figure 3) in south-western Sweden in 2019 and 2020. The site is a part of SITES[1] Skogaryd research catchment and a candidate to be a class 2 ecosystem site within the ICOS[2] research infrastructure (Heiskanen et al., 2021a). Mycklemossen mire lies within the hemiboreal forest zone. The annual 30-year average air temperature from a nearby weather stations is 6.8°C (1981-2010, SMHI Vänersborg)

and annual precipitation is 800-1000 mm (1981-2010, SHMI Vänersborg and Uddevalla). The mire is a poor fen with bog characteristics in its vegetation and pH of 3.9-4.0 (Rinne et al., 2020).

---

[1] Swedish Infrastructure for Ecosystem Science, https://www.fieldsites.se/
[2] Integrated Carbon Observation System, https://www.icos-cp.eu/

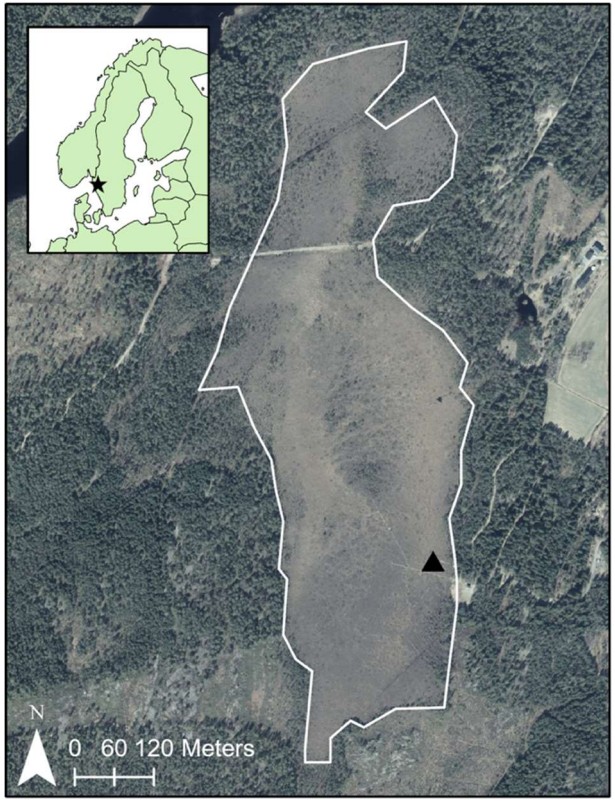

*Figure 3: Map of Mycklemossen (outlined in white). Black star indicates the location of Mycklemossen within Scandinavia, black triangle indicates the location of the chamber and NBLA measurements. Data sources: © Lantmäteriet, © EuroGeographics.*

A range of meteorological and hydrological parameters are available from the Mycklemossen research site, including air temperature, peat temperature at different depths at four locations, and water table position at three locations.

## 3.2 CH$_4$ emission and δ$^{13}$C measurements

We used two approaches to measure the δ$^{13}$C value of the emitted CH$_4$, the automated static chamber approach (e.g. McCalley et al., 2014) and the nocturnal boundary-layer accumulation (NBLA) approach (e.g. Sriskantharajah et al., 2012). With the former we obtain CH$_4$ emission rate and its δ$^{13}$C value resolved at the microtopographic scale, while with the latter we obtain an average δ$^{13}$C value of the emitted CH$_4$ over a larger area of the mire.

For the chamber approach, we used six automated chambers with dimensions of 44.5 x 44.5 x 40.5 cm. In addition, the frame onto which the collar is placed introduces additional volume, as it is approximately 5 cm high from the peat surface. This volume is more challenging to determine accurately due to the uneven peat surface. The chambers were transparent,

made out of polymethyl methacrylate, and equipped with a lid that opened and closed automatically. Each chamber was

equipped with a fan to ensure sufficient mixing of air in the chamber headspace, a soil thermometer (probe 107, Campbell Scientific, Inc., UT, USA), a PAR sensor (SQ-500, Apogee Instruments Inc., UT, USA) situated inside the chamber and a vent-tube to prevent pressure changes when opening and closing the lid. Each chamber cycle was 30 minutes and started with 5 minutes where the chamber and the tubing to and from the gas analyzer was ventilated. The chamber lid then closed for 25 minutes. The long closure time was needed to ensure a robust fit using the Keeling plot approach (Keeling, 1958). All

measurements of the methane mixing ratios and $\delta^{13}C$ were performed using a Picarro G2201-i cavity ring-down spectroscopic (CRDS) analyzer (Picarro Inc., CA, USA). The chamber measurements were conducted between 07:00 – 19:00, resulting in four measurements from each chamber every day. The time between 19:00 and 07:00 was used for measurements with the NBLA approach.

The chambers were placed along a boardwalk (Figure 4). The topography of the mire is not very pronounced with the maximum difference in surface height between chamber locations being 17 cm. Furthermore, the relative elevations were not indicative for the dominant vegetation in the chambers (Table 1, Figure 5). The vegetation in the chambers falls into three categories. In chambers 1 and 2 there is a major presence of aerenchymatous sedges, typical for moist conditions in the mire. Chamber 3 is dominated by *Sphagnum* mosses, also common in moist conditions. In chambers 4 and 5 there is considerable

presence of woody shrubs, typical for drier conditions. The vegetation in chamber 6 is intermediate between sedge-dominated and shrub-dominated.

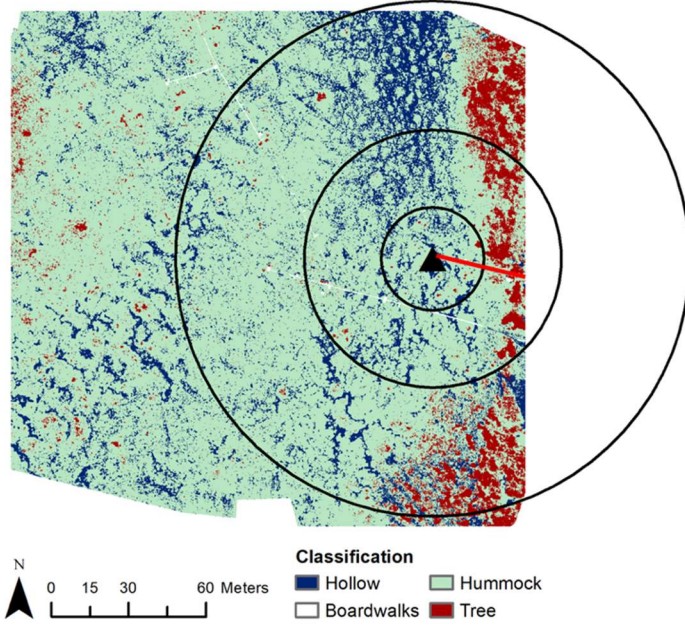

*Figure 4: Distribution of dry and wet areas in Mycklemossen according to microtopography. The black triangle indicated the sampling location of measurements used for nocturnal boundary-layer accumulation (NBLA) approach. The chambers were situated along the boardwalk (red line). Black circles indicate the distances (20 m, 50 m, 100 m) from NBLA sampling point.*

The emission rate of $CH_4$ was calculated as linear fit of $CH_4$ mixing ratio to time during the first 4 minutes of the closure. The first 60 seconds were discarded to avoid the disturbances at lid closure, leaving three minutes of data for the linear fitting. For data quality assurance $r^2$ and root-mean-square-error (RMSE) were calculated for each chamber closure. Processing and analysis of stable isotope data was conducted with MatLab (R2015b).

The $\delta^{13}C$ of the emitted $CH_4$ was obtained by the Keeling plot approach (Keeling, 1958). In this approach, we plotted the measured $\delta^{13}C$ against the inverse of the $CH_4$ mixing ratio ($\chi$). The $\delta^{13}C$ of the emitted methane was then obtained as the intercept of the $\delta^{13}C$ value at $1/\chi = 0$, by fitting a line

$$\delta^{13}C(\chi) = a + b\chi^{-1}, \tag{1}$$

to the data. Here $\delta^{13}C(\chi)$ is the observed $\delta^{13}C$ value of $CH_4$ in the chamber air at the methane mixing ratio of $\chi$, and $a$ and $b$ are coefficients obtained by line fitting. Coefficient $a$ is the intercept, which will give us the isotopic signature ($\delta^{13}C$ value) of the emitted methane. The confidence interval of the $\delta^{13}C$ at intercept was obtained by the function linfitxy in MatLab (Browaeys 2021). We removed the data from closures where the uncertainty of $\delta^{13}C$ of emitted $CH_4$ was larger than 20 ‰.

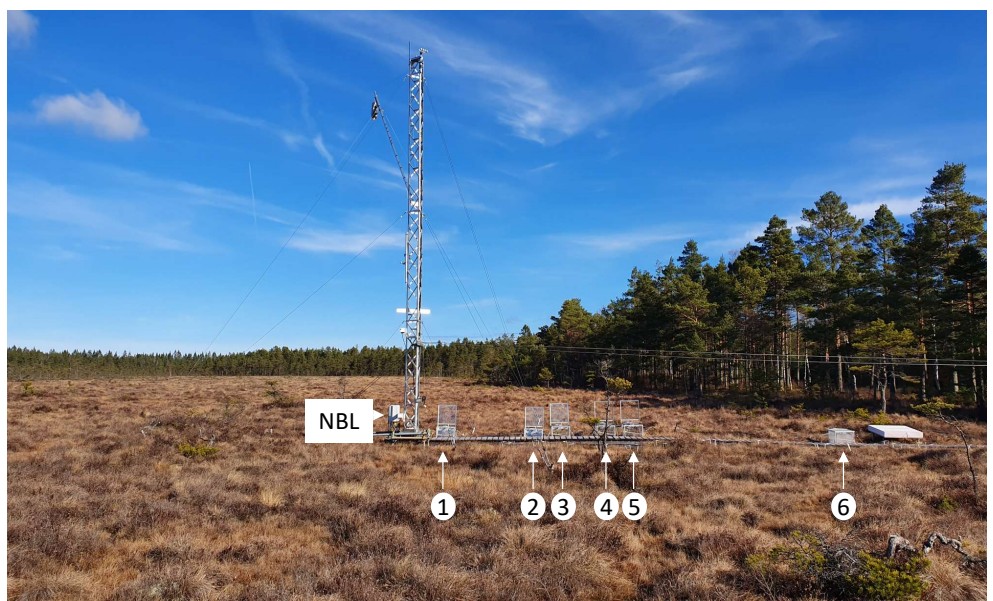

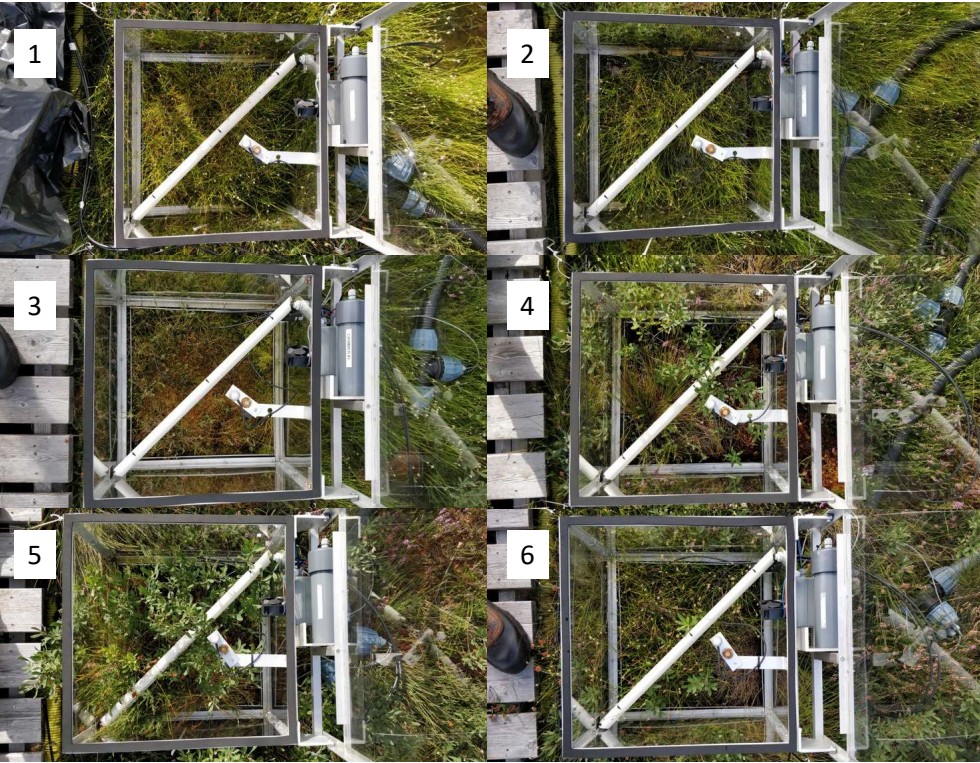

*Figure 5: Top panel: Photo showing the relative location of chambers along the boardwalk. Lower panel: Photos of vegetation inside each chamber, numbered 1-6. NBL indicates the inlet for measurement of ambient air for the nocturnal boundary-layer accumulation approach.*

For the NBLA approach we measured the CH$_4$ mixing ratio and $\delta^{13}$C at 0.4 m above the mire surface during night-time. As the emitted CH$_4$ is accumulated in the shallow stable nocturnal surface layer, we can employ a similar two-end-member mixing model as for the chamber measurements (Rinne et al., 2021). Thus, we obtain the $\delta^{13}$C of the emitted CH$_4$ by the Keeling plot approach (Eq. 1).

In addition to the automated measurements, we occasionally took manual air samples from chambers during closures and analyzed these with isotope ratio mass spectrometer, for comparison with the automated measurements. From each chamber closure, eight samples were taken into two liter Supel$^{TM}$ Inert Foil Gas Sampling bags (Sigma Aldrich, Co, LLC, USA). The eight samples from each chamber were divided into two sets, one transported to Utrecht University and the other one to Royal Holloway, University of London for analysis. The analysis methods are described by Röckmann et al. (2016) and Fisher et al. (2006). These results were compared with CRDS results and the difference in the resulting $\delta^{13}$C-CH$_4$ values of 3.4‰ was added to the $\delta^{13}$C-CH$_4$ values calculated using the CRDS data.

In order to reduce measurement noise, especially in the $\delta^{13}$C values, we aggregated the calculated CH$_4$ emissions and their $\delta^{13}$C values to ten-day averages. To analyze the spatial variability, we plotted the $\delta^{13}$C values against CH$_4$ emission rates during each ten-day interval. For the analysis of temporal variation, we plotted the $\delta^{13}$C values against the CH$_4$ emission rates from each chamber.

## 3.3 Upscaling the $\delta^{13}$C estimates

To scale up the $\delta^{13}$C values obtained from the different surface types by the chamber method to the isotopic signature of the whole mire, $\delta^{13}$C$_{mire}$, we weighted the $\delta^{13}$C values of different surface types by the areal contribution of these surface types, and by their CH$_4$ emission rates,

$$\delta^{13}C_{mire} = \left(\sum \delta^{13} C_i f_i F_i\right)\left(\sum f_i F_i\right)^{-1}, \tag{2}$$

where $\delta^{13}C_i$ is the isotopic signature of the CH$_4$ emission from the surface type $i$, $f_i$ is fraction of the mire covered by surface type $i$, and $F_i$ is CH$_4$ emission rate of the surface type $i$. Both the $\delta^{13}C_i$ and $F_i$ are based on the chamber measurements.

The map of mire surface types used to determine $f_i$ in Equation 2 was based on RGB and multispectral images collected with an Unmanned Aerial Vehicle in 2017. A random forest classifier (Breiman 2001) was used to divide the mire into three vegetation classes: hummocks, hollows and trees; producing a total accuracy of 81% (see Figure 4 and Kelly et al. 2021 for more details). Table 2 shows the proportion of each surface type for different radii around the NBLA tower. In the upscaling, average $\delta^{13}$C and CH$_4$ emission rate from chambers 1 and 2 represented the values of wet hollows while average values from

chambers 4-6 represented those from dryer hummocks. As there were very little data from chamber 3 especially in 2020, we did not use chamber 3 for upscaling. The hollows were given areal coverage of 20% and hummocks 80%.

## 3.4 Genomic analysis

Peat material for genomic analysis was collected in 2018 from three different surface types specified through the wetness classification (n = 17). Using a 1.5m long box corer, peat material was cut from the oxic-anoxic interface (~5cm) and the anoxic zone (~30cm). The peat material was immediately frozen using liquid nitrogen and stored in a -80°C freezer prior to

beginning gDNA extraction. The genetic DNA (gDNA) was extracted from 0.25 mg of peat following the DNeasy® PowerSoil® Kit manufacturer's protocol (Qiagen, Hilden, Germany).

The extracted gDNA was hybridized to a set of custom designed oligonucleotide probes which enrich the gene sequences related to $CH_4$ metabolism. This was achieved using the "captured metagenomics" method. Briefly, genes encoding enzymes

related to the $CH_4$ production and consumption were identified in the Kyoto Encyclopedia of Genes and Genomes database (KEGG) (Kanehisa et al., 2015) and were downloaded via a custom R script (https://github.com/dagahren/metagenomic-project). The target sequences downloaded from KEGG were used to design custom hybridisation-based probes for sequence capture based on the MetCap pipeline (Kushwaha et al., 2015). For further details on probe design, library construction and sequencing refer to White et al. (2022).

Libraries were multiplexed in pools of 15 in equimolar amounts based on the concentrations and sizes of samples. 1 μg of each pool was transferred to a capture tube where target gDNA was hybridised to the custom probes according to the NimbleGen SeqCap EZ SR User's Guide (Version 4.3, October 2014). The captured libraries were sequenced on an Illumina HiSeq4000 platform using sequencing by synthesis technology to generate 2 x 150 base pair paired-end reads.

Following sequencing, raw fastq files were trimmed for the presence of Illumina adapter sequences using Cutadapt version 1.2.1 (Martin, 2011). The reads were further trimmed using Sickle version 1.200 with a minimum window quality score of 20 (Joshi, 2011). The sequence reads from each of the captured data set were submitted to MG-RAST, an online metagenomic annotation program using default parameters (Meyer et al., 2008). The taxonomic abundances were annotated

using the RefSeq database (O'Leary et al., 2016) Following annotation, taxa were filtered for off-target sequences leaving only abundances of methanogenic and methanotroph microbial communities using the built in taxonomic filter within MG-RAST analysis page.

The relative abundance of methanogens and methanotrophs was calculated via the phyloseq package v1.3.0 (McMurdie and

Holmes, 2013). To allow for the small samples size and uneven distribution of replicates, a PERMANOVA was used with

999 permutations (Anderson, 2001) to identify significant differences between categories. Following square root transformation, we calculated ordination using Bray-Curtis distances and finally, a Wilkson pairwise post-hoc test was used to identify significant differences between the different wetness categories via the vegan package v2.5 (Oksanen et al., 2019). All genetic analysis was completed in R statistics package v 3.6.1 (R Core Team, 2018) and visualized using the ggplot2 package v 3.3.2 (Villanueva and Chen, 2019).

## 4 Results

### 4.1 Climate

The average daily air temperatures at the mire range from slightly below zero to above 20°C (Figure 6). Water table is typically drawn down during early summer, before being replenished by late summer and autumn rains (Figure 6). In 2018, the mire was affected by a severe heatwave and drought, as shown by the long duration of the water table drawdown, as well as from the high air temperatures that summer. The years 2019 and 2020, during which the measurements reported here were conducted, we closer to average conditions.

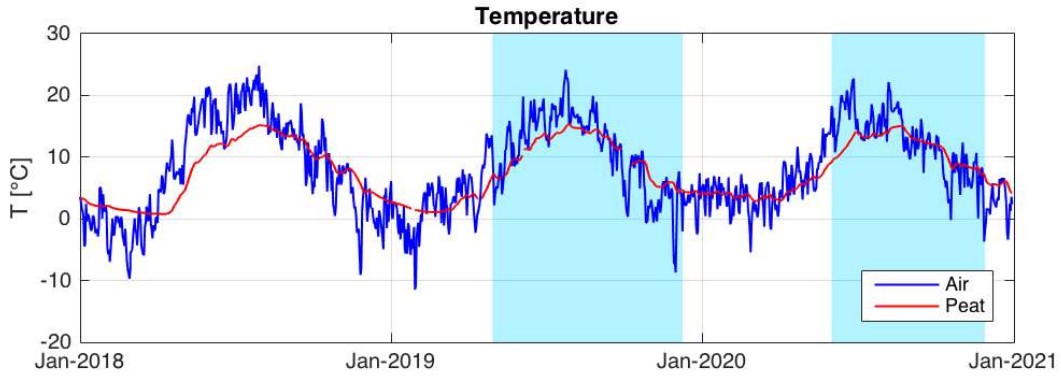

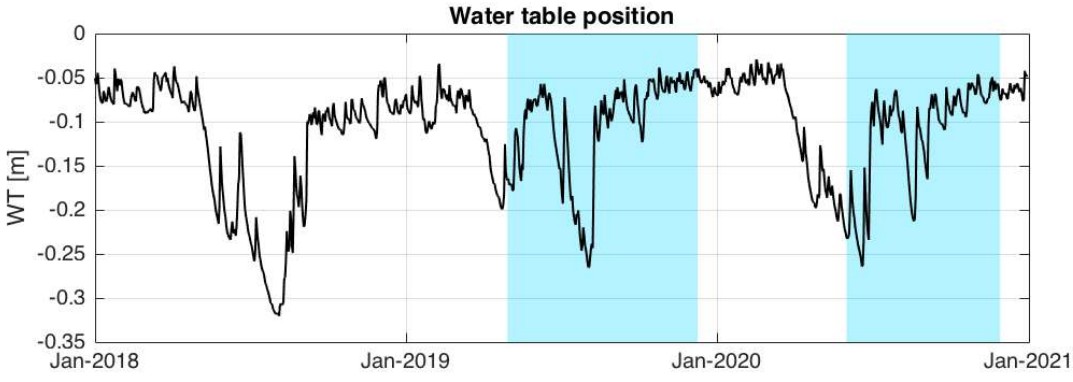

*Figure 6: Meteorological conditions during 2018-2020. Periods of $\delta^{13}C$-$CH_4$ and $F_{CH4}$ measurements are indicated by blue shading.*


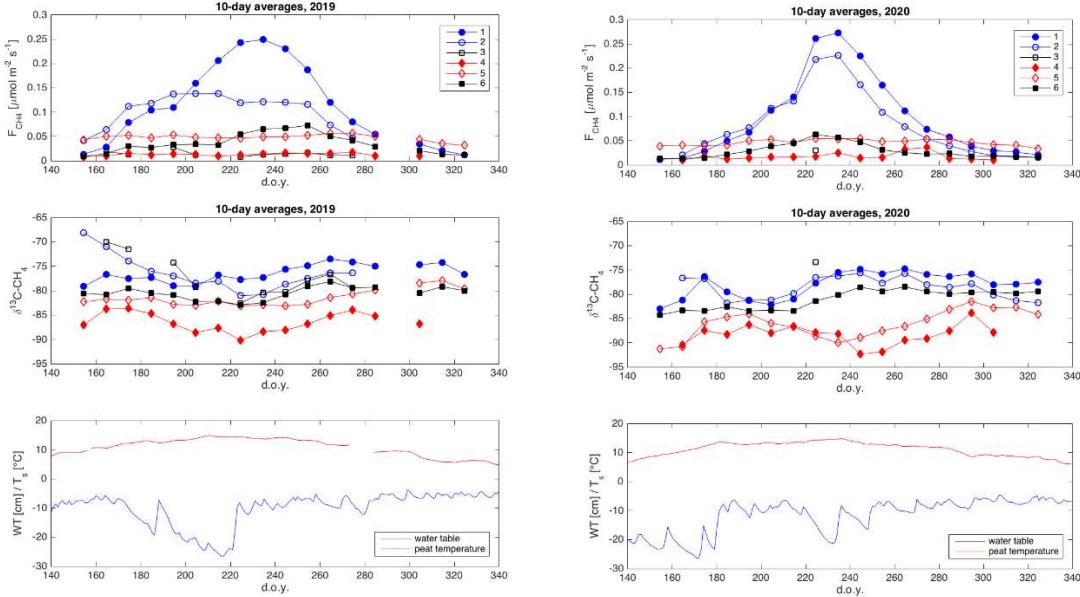

*Figure 7: Time series of ten-day averages of methane emission and $\delta^{13}C$-$CH_4$ measured from the six chambers, and peat temperature at 30 cm depth and water table position in 2019 and 2020.*

## 4.2 CH$_4$ emission rates and $\delta^{13}C$ values

The time series of CH$_4$ emission rates from most chamber locations shows a typical seasonal cycle of CH$_4$ emission, with the highest emission rates in late summer (Figure 7; Supplementary material Figure S1). We see also distinct differences between the emission rates from different chambers indicating strong small-scale spatial variation in CH$_4$ emission rate. The highest emission rates are observed from chambers 1 and 2, with abundant aerenchymatous sedges. Chambers 3 and 4 have very low CH$_4$ emission rates, despite differences in vegetation, while 5 and 6 have intermediate emission rates. Emission rate

from chamber 5 has a less pronounced annual cycle than from the other chambers.

The $\delta^{13}C$ values of emitted CH$_4$ also show relatively large differences depending on chamber location (Figure 7, Supplementary material Figure S2). In general, chamber locations with high emission rates have less depleted (less negative) $\delta^{13}C$ values of emitted CH$_4$. The seasonal cycle of the $\delta^{13}C$ values is much less obvious or systematic than that of the CH$_4$

emission rate.

The $\delta^{13}C$ values and CH$_4$ emission rates generally show a positive spatial relationship during many of the 10-day periods (Figure 8, A1 and A2). The positive relationship was more pronounced during the period of high emission rates (doy 200-

260), and more evident in 2019 than in 2020. However, chamber 3 deviated consistently during 2019 from the general behavior of the other chambers. Unfortunately, there were hardly any data that passed the quality assurance and control criteria from that chamber during 2020 due to low $CH_4$ emission rates. Omitting the data from chamber 3 led to statistically significant correlations between $CH_4$ emission rate and its $\delta^{13}C$ value during many of the 10-day periods (Fig. 8).

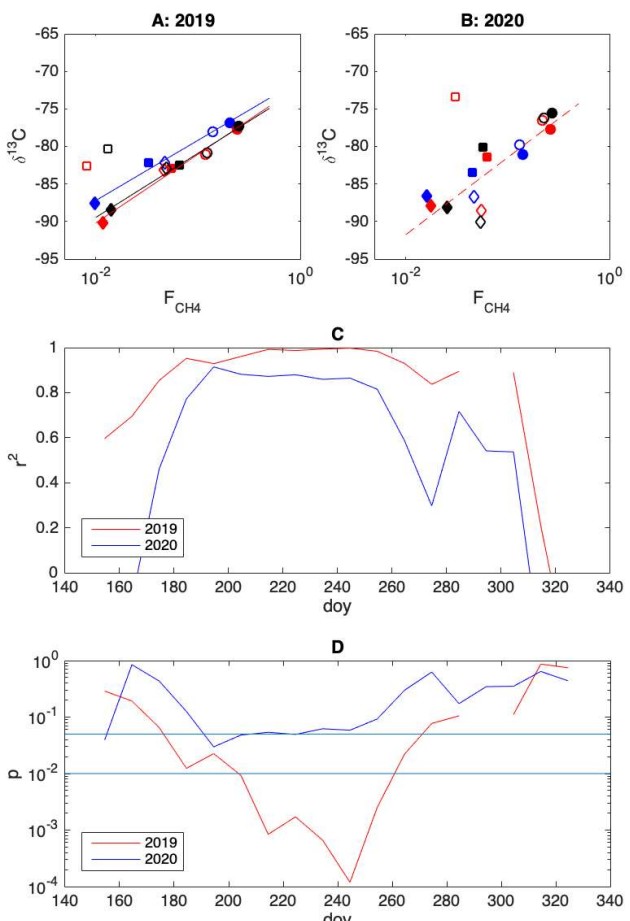

Figure 8: Panels A and B: Examples of spatial variation of ten-day averages of $\delta^{13}C$-$CH_4$ against $F_{CH4}$, during three ten-day time periods in 2019 and 2020. Chambers: 1, solid circles; 2, open circles; 3: open squares; 4: solid diamonds; 5: open diamonds; 6: solid squares. Colors of markers and lines indicate period: day of year (doy) 210-219, red; doy 220-229, blue; doy 230-239, black. Solid line indicates correlation with $p<0.01$, dashed line with $p<0.05$. Panel C: $r^2$ between ten-day averages of $\delta^{13}C$ and $F_{CH4}$, without chamber 3. Panel D: p-value of correlation, without chamber 3.

The temporal relation of $\delta^{13}C$ values and $CH_4$ emission rates showed a hysteresis-like behavior at three of the measurement locations (chambers 1, 2 and 6) during 2020 and at two locations (chambers 1 and 6) in 2019 (Figure 9). These locations are either wet or intermediate sites with relatively high emission rates. In these locations, the $\delta^{13}C$ values of emitted $CH_4$ were lower in the early part of the growing season than during a period with similar emission rates later in the season. The dry sites with lower $CH_4$ emission did not show observable systematic behavior in their $\delta^{13}C$ - $CH_4$ emission rate relation.


The $\delta^{13}C$ values of emitted $CH_4$ derived by the nocturnal boundary layer method are in the same range as the $\delta^{13}C$ values observed at the wet and intermediate chambers, with some similarities in their seasonal cycle (Figure 10, S3). The upscaling of the chamber data using the microtopographic map resulted in an average $\delta^{13}C$ value of emitted $CH_4$ of -81.3 ‰. The average $\delta^{13}C$ value of emitted $CH_4$ according to NBLA measurements was -79.3 ‰.


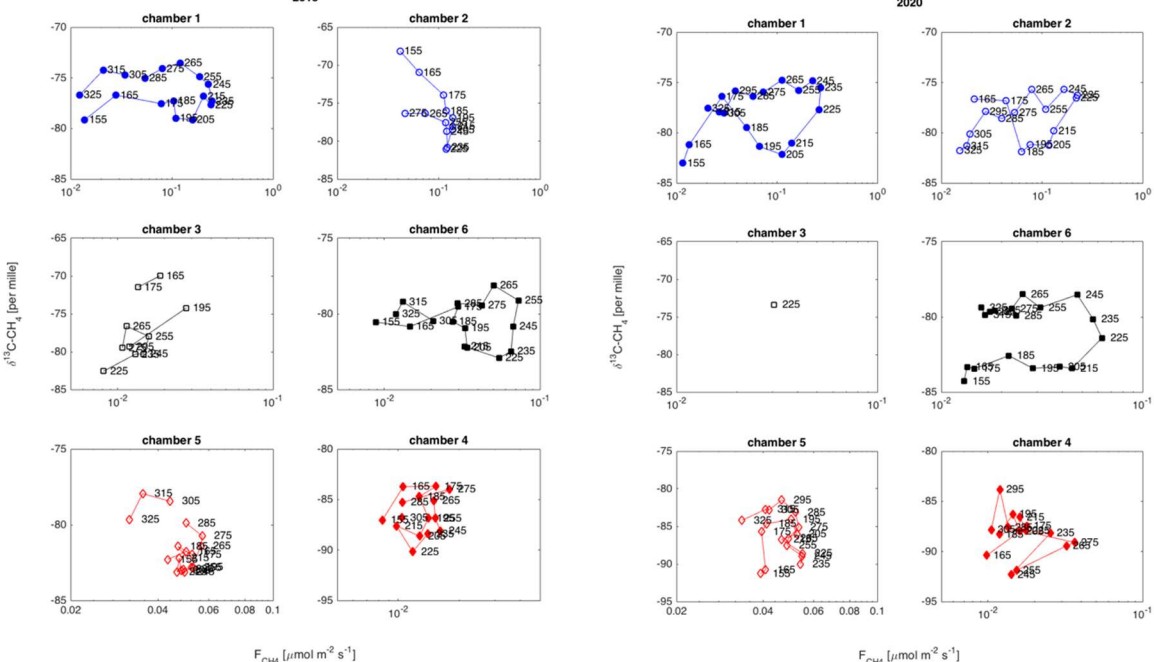

Figure 9: Temporal variation of $\delta^{13}C$-$CH_4$ against $F_{CH4}$, in each chamber location in 2019 and 2020. The marker labels indicate the day of year. Only very few data points in from chamber 3 passed the quality criteria in 2020, resulting in only one ten-day average.

## 4.3 Genomic analysis

In total, 20 methanogens and five methanotrophs were identified at *genus* level. *Genera* were spread across four classes of methanogens including Methanobacteria, Methanococci, Methanomicrobia and Methanopyri. In addition, three classes of methanotrophs including type I Gammaproteobacteria, type II Alphaproteobacteria and Verrucomicrobia were also detected. These *genera* included methanogens with the ability to perform methanogenesis via all metabolic pathways including hydrogenotrophic, acetoclastic, methylotrophic and the specialist methanogen, *Methanosarcina* (Hydr/Methyl/Aceto methanogen), which holds the ability to metabolize via multiple alternative pathways.

The proportion of methanogens to methanotrophs is a 58% to 42% split when combining all the samples. The dominant methanogens were hydrogenotrophic methanogens (46%), followed by the multiple metabolic pathway genus *Methanosarcina* (10%), with the methylotrophic and acetoclastic methanogens contributing 2% and ≤ 1% respectively. The dominant methanotrophs were the type II Alphaproteobacteria (30%), followed by type I Gammaproteobacteria (8%) and Verrucomicrobia (4%).

Significant variation in the relative abundance of taxa was observed between the wet, intermediate and dry categories ($p \leq$ 0.02) (figure 11). The PERMANOVA indicated that 37% of the variation in taxa was explained by the wetness category ($R^2$ 0.37, $p \leq 0.02$). When testing pairwise between categories, significant differences occurred between wet - dry ($p \leq 0.04$) and wet - intermediate categories ($p \leq 0.04$), but not between the dry - intermediate categories ($p \geq 0.05$).

The functional group contributing the most to dissimilarity in all comparisons was the hydrogenotrophic methanogens, with an average dissimilarity of 0.29 ± 0.19 SD between intermediate - wet, 0.20 ± 0.16 SD between intermediate - dry and finally, 0.30 ± 0.17 SD between wet - dry categories (Tables 3, 4, 5). Although contributing the highest to dissimilarity the difference was identified as non-significant when comparing between categories. Type II methanotrophs, multiple metabolic pathway Methanosarcina, Type I methanotrophs and hydrogenotrophic methanogens contributed second, third, fourth and fifth to dissimilarity, respectively. Interestingly, methylotrophic methanogens contributed little to dissimilarity but were the only methanogenic functional group to be significantly higher in abundance in wet locations when compared to intermediate ($p \leq 0.027$) and dry plots ($p \leq 0.046$). Type I methanotrophs and Verrucomicrobia methanotrophs had significantly higher average abundance in wet locations when compared to intermediate ($p \leq 0.01$) and dry plots ($p \leq 0.004$). However, type II methanotrophs were only significantly higher in abundance in wet plots when compared to dry ($p \leq 0.036$).

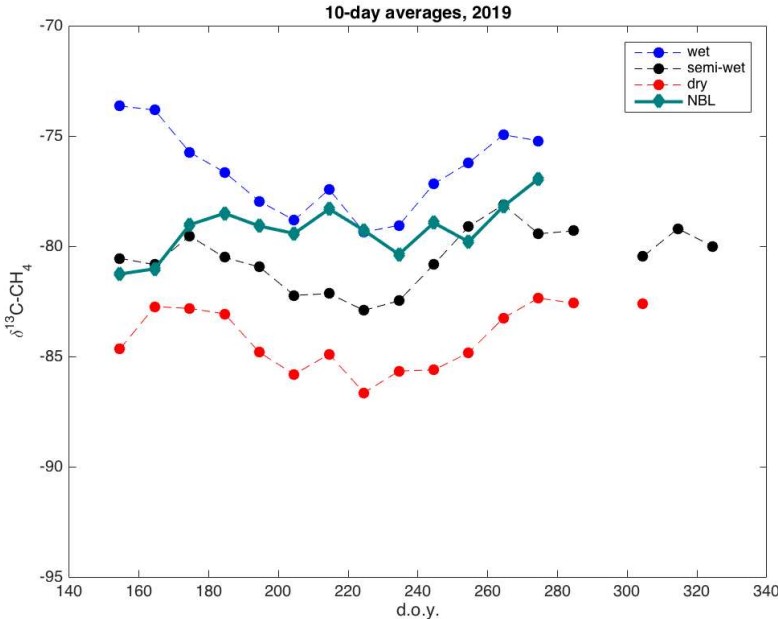

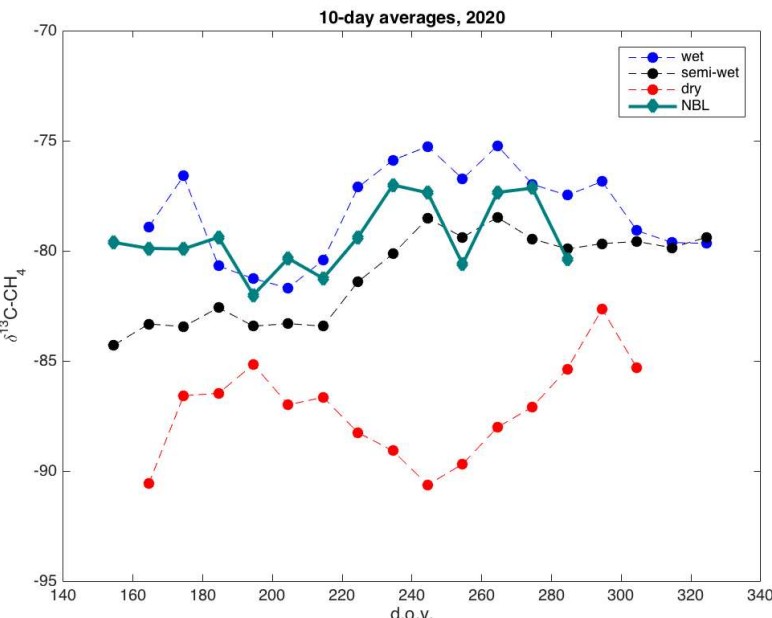

*Figure 10: Time series of ten-day average $\delta^{13}C$-$CH_4$ derived by nocturnal boundary-layer Keeling plot approach (green), and averages of wet (blue), intermediate (black) and dry (red) locations, for 2019 and 2020.*

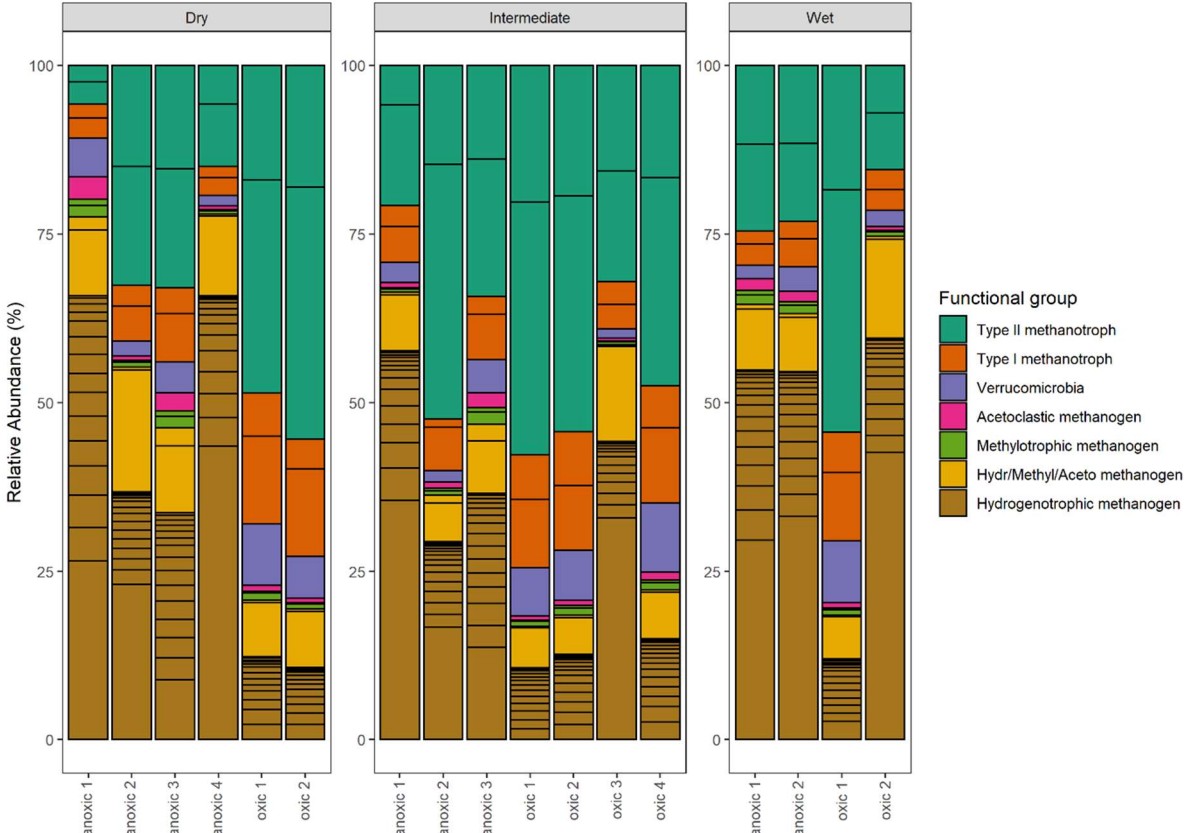

*Figure 11: Taxonomic composition: The relative abundance (%) of methanogenic and methanotrophic microbes at genus level. Color indicates functional group and which metabolic pathway is utilized during metabolism.*

## 5 Discussion

The CH₄ emitted from surfaces covered by different vegetation types show large differences in its $\delta^{13}C$ values. In the late summer of 2020, the differences between the 10-day average $\delta^{13}C$ values from different chambers were up to 10-15 ‰. Considering the modest microtopography of Mycklemossen mire, and the closeness of the measurement locations (Table 1, Figure 10), this indicates a considerable small-scale spatial variation in the processes leading to CH₄ emission. Our findings are in line with the large observed differences in CH₄ emission rates due to small-scale spatial variability from other mire ecosystems (e.g. Riutta et al., 2007; Keane et al., 2021). The spatial variation of $\delta^{13}C$ values observed at Mycklemossen are in the same range with that observed at Abisko-Stordalen mire (68°20'N, 19°30'E) in Northern Sweden by McCalley et al.

(2014). Furthermore, McCalley et al. (2014) and Mondav et al. (2017) identified the same genera of hydrogenotrophic methanogens in Abisko-Stordalen mire which we found at Mycklemossen, with the same genus (*Methanoregula*) being dominant. The similar range of the $\delta^{13}C$ values and similar methanogens at Mycklemossen and Abisko-Stordalen mires is interesting as these mires are located over 1100 km apart and differ considerably in their microtopography and climate. The microtopographic height differences at Abisko-Stordalen are about a meter, as compared to about 20 cm at Mycklemossen. Furthermore, due to the cold climate and thin wintertime snow cover Abisko-Stordalen features discontinuous permafrost in the form of palsas, whereas Mycklemossen is a hemiboreal non-permafrost mire.

The spatial variation in the $\delta^{13}C$ values of emitted $CH_4$ is systematic over the growing season and two years of measurements. Generally, the wet sedge-dominated plots with higher emission rates are associated with higher $\delta^{13}C$ values, and the dry shrub-dominated plots with lower emission rates with lower $\delta^{13}C$ values, indicating the likely importance of substrate availability and methanogenesis in determining the spatial variation in the $CH_4$ emission rate. Similar spatial relations between $\delta^{13}C$ and $CH_4$ emission rate have been observed by e.g. Hornibrook and Bowes (2007) in Welsh mires, and by McCalley et al. (2014) in a Swedish subarctic Abisko-Stordalen mire. However, the position of the chamber 3 in the $\delta^{13}C$ - $CH_4$ emission rate diagram (Fig 8, A1, A2), suggests an effect of methanotrophy on $CH_4$ emission and its $\delta^{13}C$ value from this location. This may be due to the dominance of Spaghnum mosses in this chamber, which have been shown to support considerable methanotrophy (Larmola et al., 2010). The significantly higher abundance of type II methanotrophs in wetter locations as compared to dry and intermediate supports this suggestion.

Of our two hypotheses on the origins of the spatial variation of $CH_4$ emission rates, one (HS1) assumes methanotrophy to be the key explanatory process while the other (HS2) assumes substrate availability to drive the spatial variation. The relation between the $CH_4$ emission rate and $\delta^{13}C$ values of emitted $CH_4$ we observed, especially at locations with vascular vegetation cover, mostly corroborates the latter hypothesis (HS2). Corroboration of the HS1 hypothesis would have required a negative correlation between the $\delta^{13}C$ and $CH_4$ emission rate. Furthermore, the presence of hydrogenotrophic, acetoclastic and methylotrophic methanogens enables the community to utilize all substrates available. Thus, it is unlikely that methanotrophy plays a major role in explaining the spatial variation of $CH_4$ emission from this mire system, especially as the moss dominated areas seem to cover a minor area of the mire.

As it is possible that there are seasonal differences in the factors affecting the spatial variability of the methane emission (temperature, substrate availability, methanotrophy), we analyzed the spatial variation throughout the growing seasons as ten-day averages. According to the observed spatial relations between $\delta^{13}C$ and $CH_4$ emission rates during these two growing seasons there were no major temporal shifts in the behavior of the $\delta^{13}C$ - $CH_4$ emission rate relationship (Figures A1 and A2). The $\delta^{13}C$ values and $CH_4$ emission rate, omitting chamber 3, showed tendency for positive correlation for most of the

growing seasons. Thus, it seems that the processes leading to the spatial variations in $CH_4$ emission are similar throughout the growing season.

The temporal variation in $\delta^{13}C$ was smaller and less systematic than its spatial variation. Interestingly, this temporal relation does not show similar systematic behavior than the spatial variation, indicating that the space-for-time analogy may not be valid in these seasonal time scales. The temporal behavior of $\delta^{13}C$ in relation to $CH_4$ emission rate shows a hysteresis-like behavior at some of the chamber plots. The hysteresis-like behavior is clear in wet or intermediate plots with high emission rates. The lack of observable hysteresis-like behavior in the other plots could be due to the small range of emission rates leading to random variation in the data to mask the any systematic behavior. The hysteresis-like behavior indicates that the temporal variation of $CH_4$ emission rates from this mire could be a result of two time-lagged compounding effects, following the HT3 hypothesis, especially as the variation of $\delta^{13}C$ value and $CH_4$ emission rate in mire-scale are mostly affected by the high-emitting surfaces. The increasing $CH_4$ emissions during the first half of the growing season could be caused by increasing peat temperature enhancing the activity of methanogenic Archaea (Juottonen et al., 2008). Later in the growing season, the increased input of root exudates from vascular plants would increase the substrate availability, resulting in higher $\delta^{13}C$ values than in the early season, yet similar $CH_4$ emission rates. However, we cannot assign the whole seasonal cycle of $CH_4$ emission rates to changes in substrate availability, as this would result in a pronounced positive relationship between $\delta^{13}C$ and $CH_4$ emission rates, which we did not observe. According to the genetic analysis, the microbial community holds the functional potential to produce $CH_4$ via the hydrogenotrophic and acetoclastic pathways, thus enabling shifts in $\delta^{13}C$ following the seasonal changes in availability of substrate. However, the highly depleted $\delta^{13}C$, mostly between -90 ‰ and -70 ‰, does indicate dominance of hydrogenotrophic methane production at this mire, as the hydrogenotrophic pathway produces $CH_4$ with $\delta^{13}C$ below 60 ‰, while acetoclastic pathway would result in $CH_4$ with $\delta^{13}C$ above -60 ‰ (Whiticar, 1999; McCalley et al., 2014). Therefore the changes in $\delta^{13}C$ of emitted $CH_4$ are most likely due to energetics of the hydrogenotrophic methanogenesis (Penning et al., 2005; Hornibrook 2009). The hysteresis between temperature and $CH_4$ emission, as observed by Chang et al., (2020; 2021) and Łakomiec et al. (2021), could be partly due to the seasonal development of peat temperature and partly due to the changes in substrate availability for methane production.

The $\delta^{13}C$ values of emitted $CH_4$ derived by the nocturnal boundary-layer approach (NBLA) corresponded in magnitude to the values of the wet and intermediate surfaces. As these surfaces dominate the emission, it is natural that the NBLA approach will correspond to these more closely than to the dry surfaces with low $CH_4$ emission. The up-scaled $\delta^{13}C$ from the chamber measurements was in a similar range to the mire-scale $\delta^{13}C$ measured by the NBLA method, indicating the dominance of hydrogenotrophic methanogenetic pathways. Obtaining reliable mire scale isotopic signatures is crucial, for example for the use of isotopic data for source apportioning of $CH_4$ by atmospheric inversions. Here we show that the chamber $\delta^{13}C$ measurements can be successfully upscaled using a mire surface characterization based on UAV data. Such an approach

enables the calculation of mire-scale $\delta^{13}C$ estimates at sites where NBLA measurements are not available. In combination with UAV-upscaled $CO_2$ fluxes (e.g. Kelly et al 2021), there are further opportunities to examine the impacts of spatial variations in vegetation productivity and respiration on $CH_4$ emission rates and $\delta^{13}C$ values.

The mire scale $\delta^{13}C$ value of emitted $CH_4$ observed at Mycklemossen (-81‰ to -79‰) is somewhat lower than observations in northern Scandinavia by Fischer et al., (2017) and in the lower end of the wetland $\delta^{13}C$-$CH_4$ distribution as presented by Menoud et al., (2022). All these show considerably lower $\delta^{13}C$ values of $CH_4$ emitted from northern mire ecosystems than the average $\delta^{13}C$ values for wetland $CH_4$ emissions used in many atmospheric inversion studies (-60‰ to -58‰; Mikaloff-Fletcher et al., 2004a,b; Bousquet et al., 2006; Monteil et al., 2011). Together with the wider data sets of Fisher et al., (2017) and Menoud et al. (2022), the observations presented here would support using a lower $\delta^{13}C$ value for $CH_4$ emitted from northern mire ecosystems in atmospheric inversion studies.

## 6 Conclusions

We conducted automatic chamber and nocturnal boundary layer (NBLA) measurements of $\delta^{13}C$ values of emitted $CH_4$, as well as genomic analyses of the $CH_4$-relevant microbial communities, to investigate the drivers of the spatial and temporal variability of $CH_4$ emission rate and $\delta^{13}C$ value in a hemiboreal Swedish mire. Despite the small elevation differences (<20 cm) between the microtopographic zones in the mire, we observed stark contrasts in the $CH_4$ emission rates and $\delta^{13}C$ values between the zones, similar in magnitude to mires which have much more pronounced microtopography. According to the relationships between $\delta^{13}C$ values and $CH_4$ emission rates we observed, the spatial variability of $CH_4$ emission from Mycklemossen mire is unlikely to be controlled mostly by methanotrophy. Instead, variations in methanogenesis due to the differences in substrate availability, following our hypothesis 2 on spatial variability (HS2), seem to be a more likely source of most of the variation in $CH_4$ emission rates. The seasonal variation of $CH_4$ emission is likely controlled by both temperature and substrate availability, leading to hysteresis-like behavior in the $\delta^{13}C$ - $CH_4$ emission rate relationship, following our hypothesis 3 on temporal variability (HT3). The taxonomic data shows the functional potential to produce $CH_4$ via multiple metabolic pathways, enabling shifts following changes the substrate availability. However, the highly depleted $\delta^{13}C$ values observed indicate the dominance of hydrogenotrophic methanogenesis, and thus the variation in $\delta^{13}C$ may be due to the energetics of this process. Interestingly, the measurement plot with Sphagnum-dominated vegetation diverged from the general spatial $\delta^{13}C$-$F_{CH4}$ relation, warranting future studies on this vegetation type. In addition, we confirmed that drone-based upscaling of $\delta^{13}C$ chamber measurements provides reliable mire-scale estimates when compared to NBLA $\delta^{13}C$ estimates. The observed mire scale $\delta^{13}C$ values were in the lower end of reported $\delta^{13}C$ values from northern mires and together with these, support the need for revising the $\delta^{13}C$ value for northern wetland systems used in atmospheric inversion studies. The results obtained can help to constrain our theories on the causes of the variability of methane emission from

mire ecosystems and can thus be useful in development of numerical models of mire biogeochemistry, needed to predict the

fate of northern mire ecosystems in the changing climate.

**Appendix A: Spatial $\delta^{13}$C-CH$_4$ - F$_{CH4}$ relations as ten-day averages**

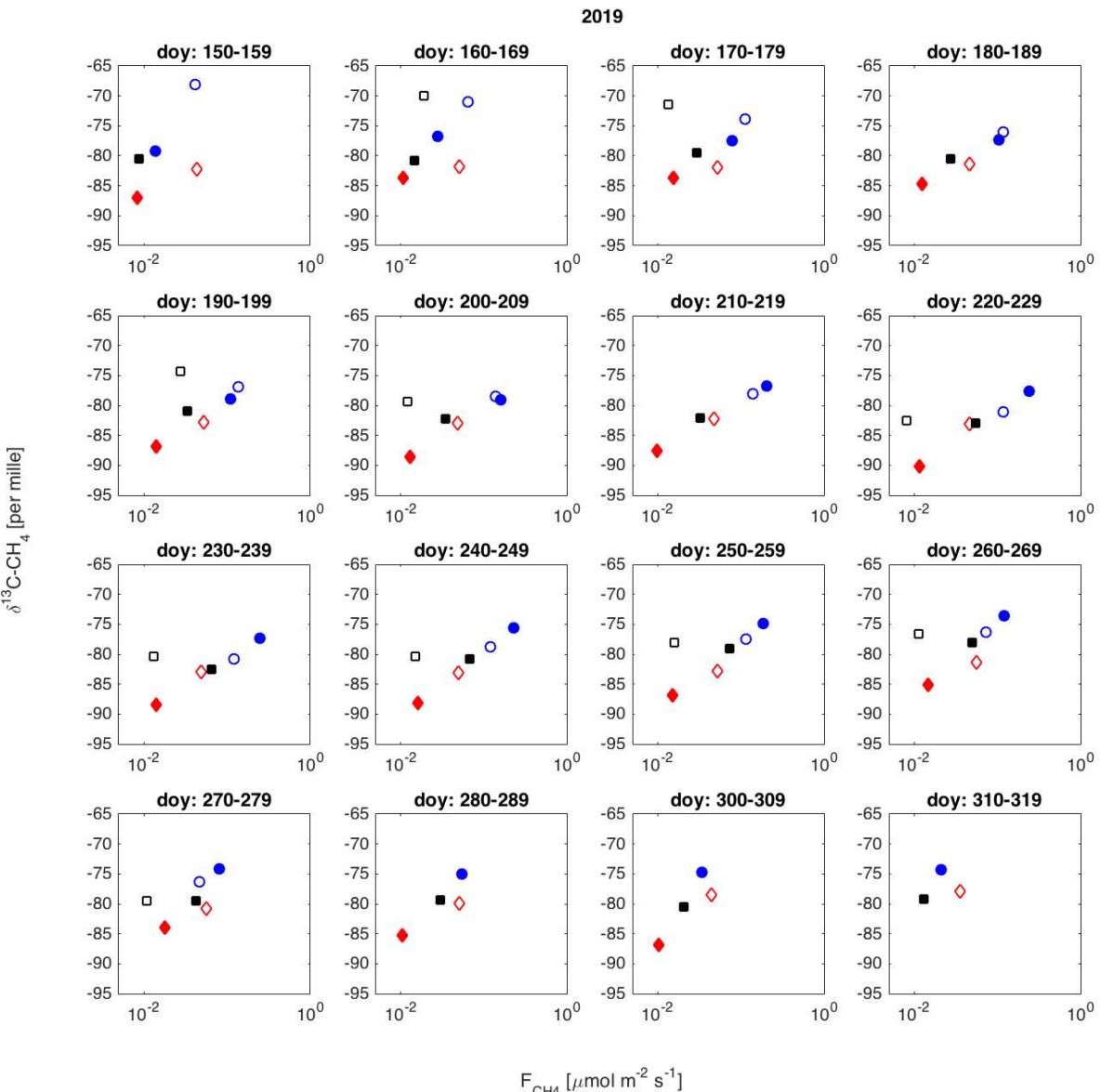

495

*Figure A1: Spatial variation of $\delta^{13}$C-CH$_4$ against F$_{CH4}$, as ten-day averages during 2019. Chamber 1: blue solid circle; 2: blue open circle; 3: black open square; 4: red solid diamond; 5: red open square; 6: black solid square.*

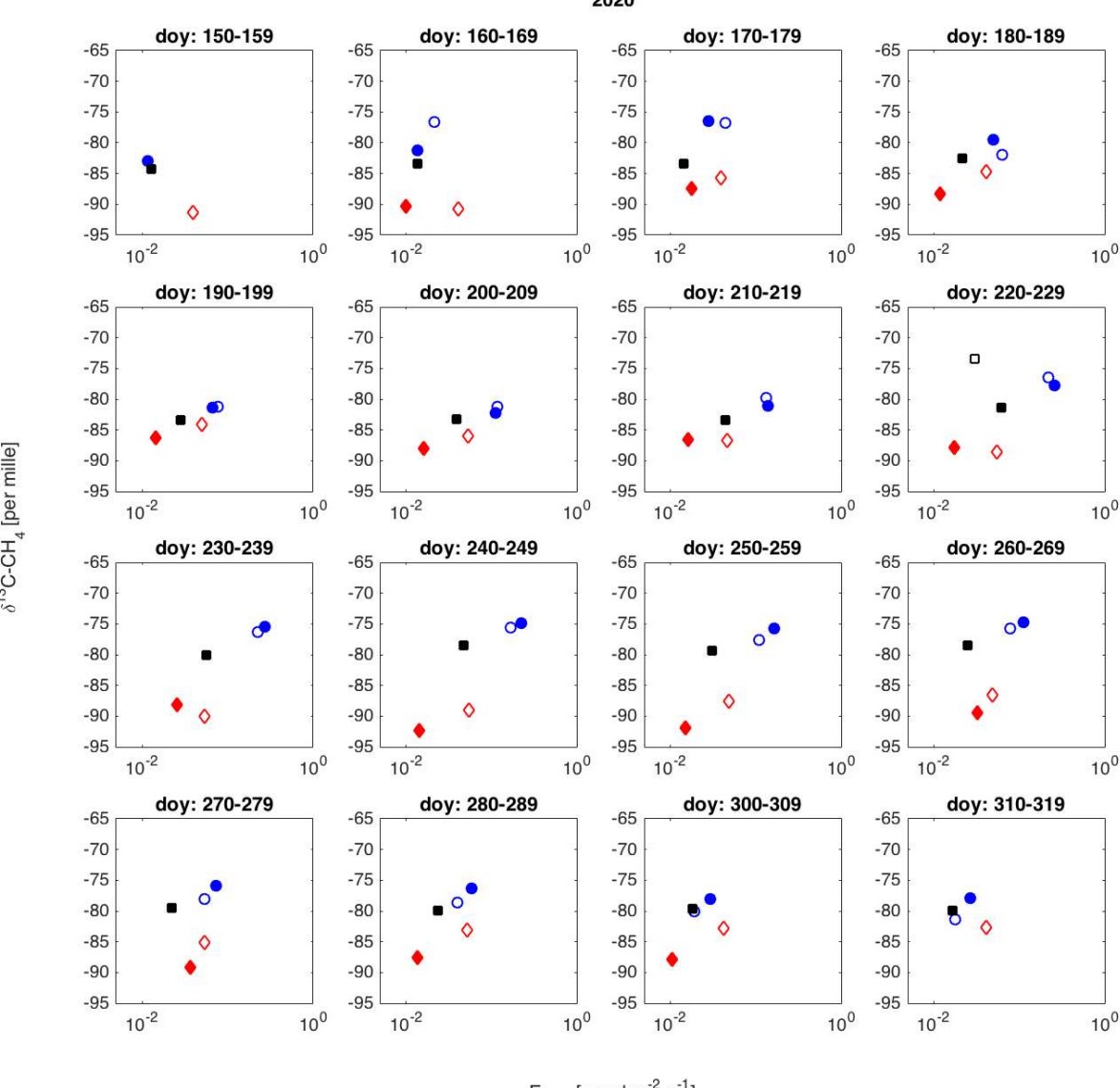

*Figure A2: Spatial variation of $\delta^{13}C\text{-}CH_4$ against $F_{CH4}$, as ten-day averages during 2020. Chamber 1: blue solid circle; 2: blue open circle; 3: black open square; 4: red solid diamond; 5: red open square; 6: black solid square.*

**Acknowledgements**

We thank Malika Menoud and Prof. Thomas Röckmann at IMAU, and Dr. David Lowry at RHUL for help in isotopic analysis of bag samples. The access to the Mycklemossen site was made possible by Swedish Infrastructure for Ecosystem Sciences (SITES, co-financed by the Swedish Research Council) and ICOS Sweden network (co-financed by the Swedish Research Council (grant no. 2015-06020, 2019-00205). Probe hybridization and sequencing was performed at the Center for Genomic Research, University of Liverpool. Data handling was enabled by resources in project (SNIC 2019/8-365) provided by the Swedish National Infrastructure for Computing (SNIC) at UPPMAX, partially funded by the Swedish Research Council through grant agreement no. 2018-05973.

**Financial Support**

This research has been supported by the MEthane goes Mobile: MEasurement and MOdeling (MEMO2) project from the European Union's Horizon 2020 research and innovation program under the Marie Skłodowska-Curie grant no. 722479, and by the Greenhouse Gas Fluxes and Earth System Feedbacks (GreenFeedBack) project from the European Union's Horizon Europe – Framework Programme for Research and Innovation (project no. 101056921). We acknowledge Crafoord foundation for financial support for financing the Picarro isotope analyzer. We acknowledge funding from Strategic Research Area BECC (2018).

**Data availability**

The annotated metagenomes are available at the MG-RAST server under the project ID: 91145. The isotopic and methane emission data is available at zenodo: 10.5281/zenodo.6385096

**Code availability**

Code used in the taxonomic analysis can be found at https://github.com/joel332/Analysis-of-captured-metagenomic-data/blob/main/Mycklemossen_isotopes_taxanomic_analysis. The code for methane flux and isotopic analysis is available at zenodo: 10.5281/zenodo.6670314.

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

**Tables**

Table 1 : Dominant vegetation in flux chambers. D : dominant ; P : present. Niche indicates the niche of the species. The relative elevation (above 80 m a.s.l.) of moss surface at each chamber is indicated.


| SPECIES | CH_1 13 CM | CH_2 6 CM | CH_3 6 CM | CH_4 14 CM | CH_5 20 CM | CH_6 3 CM | NICHE |
|---|---|---|---|---|---|---|---|
| *Rhynchospora alba* | D | D | - | - | - | D | Wet |
| *Eriophorum vaginatum* | - | - | P | D$^{50\%}$ | P | P | Wet - Moist |
| *Andromeda polifolia* | - | - | P | - | - | - | Moist |
| *Myrica gale* | - | - | - | D$^{50\%}$ | D | P | Moist |
| *Erica tetralix* | - | - | P | P | P | P | Moist |
| *Calluna vulgaris* | - | - | P | P | - | - | Moist – Dry |
| *Sphagnum papillosum* | - | - | D | P | - | - | Moist |

Table 2 : Proportions of different vegetation types in different radii around the NBLA tower.

| Radius [m] | Wet hollows [%] | Dry hummocks [%] | Trees [%] |
|---|---|---|---|

| 20 | 20 | 78 | 1.0 |
|---|---|---|---|
| 50 | 16 | 76 | 7.2 |
| 100 | 17 | 75 | 8.6 |

Table 3 : Results of SIMPER analysis between intermediate (n = 7) and wet (n = 4) plots. Functional group are ranked according to their average contribution to dissimilarity between plots. Standard deviation (SD), average abundances, percentage of cumulative contribution and permutation *p*-value (Probability of getting a larger or equal average contribution in random permutation of the group factor) are also included.

| Functional group | Average dissimilarity | SD | Average abundance intermediate | Average abundance Wet | Cumulative Percentage | p |
|---|---|---|---|---|---|---|
| Hydrogenotrophic methanogens | 0.30 | 0.19 | 3155 | 20214 | 48% | 0.10 |
| Type II methanotrophs | 0.18 | 0.13 | 3583 | 11503 | 76% | 0.08 |
| Hydr/Methyl/Aceto methanogens | 0.06 | 0.03 | 839 | 3844 | 85% | 0.13 |
| Type I methanotrophs | 0.05 | 0.04 | 821 | 3006 | 93% | **0.01** |
| Verrucomicrobia | 0.03 | 0.02 | 281 | 1482 | 97% | **0.00** |
| Methylotrophic methanogens | 0.01 | 0.01 | 94 | 715 | 99% | **0.03** |
| Acetoclastic methanogen | 0.01 | 0.01 | 80 | 605 | 100% | 0.13 |



Table 4 : Results of SIMPER analysis between intermediate (n = 7) and dry (n = 6) plots. Taxa are ranked according to their average contribution to dissimilarity between plots. Standard deviation (SD), average abundances, percentage of cumulative contribution and permutation $p$-value (Probability of getting a larger or equal average contribution in random permutation of the group factor) are also included.

| Functional Group | Average dissimilarity | SD | Average abundance intermediate | Average abundance dry | Cumulative Percentage | p |
|---|---|---|---|---|---|---|
| Hydrogenotrophic methanogens | 0.21 | 0.16 | 3155 | 4050 | 52% | 0.95 |
| Type II methanotrophs | 0.11 | 0.10 | 3583 | 2105 | 80% | 0.96 |
| Hydr/Methyl/Aceto.methanogens | 0.05 | 0.04 | 839 | 1100 | 92% | 0.77 |
| Type I methanotrophs | 0.02 | 0.01 | 821 | 676 | 97% | 0.99 |
| Acetoclastic methanogens | 0.01 | 0.01 | 80 | 104 | 98% | 0.91 |
| Methylotrophic methanogens | 0.00 | 0.00 | 94 | 110 | 99% | 0.97 |
| Verrucomicrobia | 0.00 | 0.00 | 281 | 294 | 100% | 1.00 |





Table 5 : Results of SIMPER analysis between wet (n = 4) and dry (n = 6) plots. Taxa are ranked according to their average contribution to dissimilarity between plots. Standard deviation (SD), average abundances, percentage of cumulative contribution and permutation $p$-value (Probability of getting a larger or equal average contribution in random permutation of the group factor) are also included.

| Functional Group | Average dissimilarity | SD | Average abundance wet | Average abundance dry | Cummulative Percentage | p |
|---|---|---|---|---|---|---|
| Hydrogenotrophic methanogens | 0.30 | 0.18 | 20214 | 4050 | 47.46% | 0.11 |
| Type II methanotrophs | 0.19 | 0.12 | 11503 | 2105 | 77.08% | **0.04** |
| Hydr/Methyl/Aceto methanogens | 0.05 | 0.03 | 3844 | 1100 | 85.58% | 0.39 |
| Type I methanotrophs | 0.05 | 0.04 | 3006 | 676 | 93.15% | **0.00** |
| Verrucomicrobia | 0.03 | 0.02 | 1482 | 294 | 97.20% | **0.00** |
| Methylotrophic methanogens | 0.01 | 0.01 | 715 | 110 | 98.70% | **0.05** |
| Acetoclastic methanogens | 0.01 | 0.01 | 605 | 104 | 100.00% | 0.14 |
