# Peer review of "Spatial and temporal variation of $\delta^{13}C$ values of methane emitted from a hemiboreal mire: Methanogenesis, methanotrophy, and hysteresis"

_Biogeosciences, 2022_

## Referee Comment (RC2)

Spatial and temporal variation of $^{13}$C signature of methane emitted from a temperate mire:
Methanogenesis, methanotrophy, and hysteresis

**General comments**
The study by Rinne et al. investigates $CH_4$ emission rates and $\delta^{13}C$-$CH_4$ values, and the
community structure of methanogenic and methanotrophic communities in a poor fen in
southwest Sweden. It is one the most detailed investigation to date pairing high temporal
resolution upscaled $\delta^{13}C$-$CH_4$ values with integrated $\delta^{13}C$ values of $CH_4$ flux sampled from
nocturnal boundary-layer accumulation. The key findings locally are that: (i) the observed spatial
and temporal differences in $\delta^{13}C$ values of $CH_4$ emissions vary systematically in response to
environmental conditions, (ii) the spatial range of values (~15 permil) is larger than temporal
variations and appears to be governed by differences in substrate and moisture levels within the
peatland that can be identified by vegetation assemblages that can be delineated via remote
sensing, and (iii) metagenomic analysis indicates that methanogenic communities within the
peatland are diverse and capable of adapting to changes in substrate supply and environmental
conditions. I support publication of this work with minor revision.

I recommend that the authors explore further in the Discussion section the implications of their
measured $\delta^{13}C$ values for isotope-weighted global $CH_4$ budgets. The measured $\delta^{13}C$ values (~ -
81 to -79 permil) of $CH_4$ emissions from the site are significantly more negative than $\delta^{13}C$ values
typically attributed to global and northern wetlands (e.g., -58‰; Mikaloff-Fletcher et al.,
2004a,b; -58‰, Bousquet et al., 2006; -59‰, Monteil et al., 2011;). Similar to Fisher et al.
(2017), this study presents further compelling evidence for a need to adjust $\delta^{13}C$ values attributed
to $CH_4$ emissions from northern peatlands.

**Specific comments**
Manuscript title: '…variation of $\delta^{13}C$ values of methane…'

Line 13 – '…offer clues…'?

Line 76-77 and elsewhere. Replacing terms such as 'isotopically lighter $CH_4$' with more specific
language would eliminate the need for clarifying statements in parentheses. For example (lines
75-76) could be written as ' … hydrogenotrophic methanogenesis typically produced $CH_4$ that is
$^{13}$C-depleted relative to $CH_4$ generated from acetoclastic methanogenesis.'

Line 108: 'reflect differences in $CH_4$ production due to differences in substrate availability for
methanogenesis."

Line 110 and elsewhere: 'methanotrophy prefers $^{12}$C, leaving more $^{13}$C to the emitted $CH_4$" =
'Enzymatic reactions associated with methanotroph metabolism consume $^{12}CH_4$ preferentially,
resulting in $^{13}$C-enrichment of residual $CH_4$.'

Line 113 – awkward sentence; 'less $^{13}$C depleted $CH_4$' = '$^{13}$C-enriched $CH_4$' or '$CH_4$ having
more positive $\delta^{13}C$ values'.

Line 121 – In this context 'substrate supply' rather than 'trophic status' perhaps would more accurately describe the environmental variable impacting $CH_4$ emission rates.

Line 163 – remove capitalization 'polymethyl…'

Lines 203-205 – How was the CRDS calibrated in the field for concentration and stable isotope measurements?

Lines 231-233 – Data from chamber 3 are not mentioned?

Line 276 – '…seems to be quite similar…' If this is an important point, perhaps employ a statistical comparison?

Line 304 – 'there were hardly any data'

*Ed Hornibrook*

**References**

Bousquet, P., Ciais, P., Miller, J.B., Dlugokencky, E.J., Hauglustaine, D.A., Prigent, C., Van der Werf, G.R., Peylin, P., Brunke, E.G., Carouge, C., Langenfelds, R.L., Lathiere, J., Papa, F., Ramonet, M., Schmidt, M., Steele, L.P., Tyler, S.C. and White, J. (2006) Contribution of anthropogenic and natural sources to atmospheric methane variability. Nature 443, 439-443.

Fisher, R. E., France, J. L., Lowry, D., Lanoisellé, M., Brownlow, R., Pyle, J. A., et al. (2017). Measurement of the $^{13}C$ isotopic signature of methane emissions from northern European wetlands. *Global Biogeochemical Cycles*, 31, 605–623.

Mikaloff Fletcher, S. E., Tans, P. P., Bruhwiler, L. M., Miller, J. B., and Heimann, M. (2004a). $CH_4$ sources estimated from atmospheric observations of $CH_4$ and its $^{13}C/^{12}C$ isotopic ratios: 1. Inverse modeling of source processes. *Global Biogeochem. Cy.* 18:GB4004, doi:10.1029/2004GB002223.

Mikaloff Fletcher, S. E., Tans, P. P., Bruhwiler, L. M., Miller, J. B., and Heimann, M. (2004b). $CH_4$ sources estimated from atmospheric observations of $CH_4$ and its $^{13}C/^{12}C$ isotopic ratios: 2. Inverse modeling of $CH_4$ fluxes from geographical regions. *Global Biogeochem. Cy.* 18:GB4005, doi:10.1029/2004GB002224.

Monteil, G., Houweling, S., Dlugockenky, E. J., Maenhout, G., Vaughn, B. H., White, J. W. C., and Rockmann, T. (2011). Interpreting methane variations in the past two decades using measurements of $CH_4$ mixing ratio and isotopic composition, Atmos. Chem. Phys., 11, 9141–9153.

---

## Author Response (AR1)

***General Comments***

***The authors present an interesting and valuable dataset showing temporal and spatial patterns of mire methane flux and its $^{13}C$ signature. They aimed to disentangle the relative importance of methanotrophy vs methanogenesis as well as the availability of substrates for methanogenesis for explaining temporal and spatial variability hypotheses.in their data. Secondary goals were to describe the methane associated prokaryotic community and compare the mire-level $^{13}C$ signature from upscaled measurements (using their chambers and a land cover map for the mire derived from a previous study) to nocturnal boundary layer measurements. While the data itself are useful, and the upscaled $^{13}C$ method successful, there are substantial issues. Primarily that the data presented are insufficient to fully test their hypotheses.***

We thank the reviewer for his constructive and thoughtful comments. The analysis and interpretation of our d13C data does indeed indicate that this data cannot fully solve the question between the hypotheses, as is now hopefully better conveyed in the revised manuscript. We have now reformulated our discussion to take the uncertainties and complexities better into account. However, in our opinion a conceptual framework with simplified hypotheses offers a useful guide for data analysis and interpretation.

***Spatial (HS1 and HS2):***

***HS1 proposes that variation (in methane flux and $^{13}C$ signature) is due to spatial changes in methane consumption, while HS2 proposes variation is due to spatial shifts from hydrogenotrophic to acetoclastic methane production.***

***These are not mutually exclusive, which is not inherently an issue, although they are treated as such in the study. Without information on the spatial distribution of methanotrophs or the respective groups of methanogens (or their substrates) or well-constrained values for the expected $^{13}C$ signatures from individual processes, any conclusions on their relative contribution to the data is conjecture.***

As the reviewer states, the two hypotheses on spatial variation are not mutually exclusive and they are not intended to be such, as is implied e.g. by the discussion of behavior of the data from chamber 3. However, their purpose is to act as useful simplifications to provide a conceptual framework to help analyzing and interpretation of the data. We agree that the presentation of the hypotheses should stress the non-exclusivity better, and we have modified the revised manuscript to include a case where either of the two processes of HS1 and HS2 dominates, by including a zero-hypothesis. We have also re-titled the "hypotheses" chapter as "Conceptual framework", and in general make the role of the presented hypotheses as simplifications to aid data interpretation clearer, in chapters Conceptual framework, Discussion, and Conclusions, as well as in the abstract.

We do not agree that all conclusions on the contribution of certain processes to the variation in d13C and FCH4 are just conjecture, as the data goes some way into refuting some hypotheses. What we state is that the observation of positive correlation between d13C and FCH4 does show that it is unlikely that methanotrophy would be the dominant cause of the spatial variation in FCH4. This is stated in the Discussion of the revised manuscript.

*Temporal (HT1 – HT3)*

*HT1 states that temporal variation (in methane flux and $^{13}$C signature) is driven by temperature. HT2 proposes temporal shifts from hydrogenotrophic to acetoclastic methane production. HT3 wisely combines them and proposes there will be a time lag between temperature and production of substrate (presumably of acetate) in the ecosystem that produces a hysteretic out and back arc in the data.*

*The presence of HT3 resolves the issue of exclusivity there, however the issue of being able to ascribe 13C signature changes to changes in microbial processes without any constraining values or direct measurement of those processes remains. Additionally, although the evidence from their data is evenly split between a temperature-driven response (see point clouds in Figure 9) and a response indicative of hysteresis, they conclude that HT3 is supported.*

Also here, as in temporal hypotheses, the different hypotheses are simplifications designed to aid the analysis and interpretation of the data. It is true that we do not have data on temporal development of the microbial communities. As this would have required much larger resources this is out of the scope of this study. The central aim of this study is to analyze the spatial and temporal variations of d13C and CH4 emission rate, to find out which hypotheses they may refute or corroborate.

The chambers from which no hysteretic behavior of d13C-FCH4 relation is observed are those with low methane emission. Thus, the random uncertainty of the measurements leads to a larger relative noise in the data, which can mask any relatively hysteretic behavior in these chambers. This is now stated in the Discussion chapter of the revised manuscript. The hysteretic behavior is evident in the high-flux locations which dominate the mire-scale CH4 emission, and its d13C value. This interpretation is also stated in the Discussion of the revised manuscript.

We have revised our reasoning on the roles of the hydrogenotrophic and acetoclastic methanogenesis. We now interpret the data to indicate hydrogenotrophic methanogenesis to be likely due to the very depleted 13C/12C ratio. The energetics of this methanogenetic pathway, due to changes in substrate availability, can be controlling the d13C variations. We have modified the Discussion and Conclusions of the revised manuscript to reflect this interpretation.

*While the dataset is strong, its strength is mainly in describing spatial and temporal variation in methane flux and signature. If data are available from the site on the $^{13}$C signature of soil C, this might be enough to draw conclusions on microbial processes*

*based on assumptions regarding fractionation rates. Otherwise, the framing of the study should shift to focus on its strengths; I can imagine an analysis of hotspots and hot moments, and/or the relationship between fluxes, vegetative cover, and water depth.*

This description of spatial and temporal variation of d13C and its covariation with CH4 emission rate are the central themes of this study. The different hypotheses, presented in the chapter on Conceptual framework of the revised manuscript, are to be treated as useful simplifications to be used as framework for data interpretation. This is emphasized on the revised version of the manuscript.

If "hotspots" and "hot moments" are to mean times and locations with considerably higher emission rate that in the near-by spatio-temporal environment, we do not really observe such events or locations (see Figures S1 and S2). What we observe is more like a continuum, with gradual changes overlaid by some variation.

We feel that an analysis of CH4 emission rates in relation with vegetation cover and water depth would not be very novel as it has been conducted in many previous studies. Also, this data set has rather limited spatial coverage (six chambers) due to the instrumental requirement of isotope Keeling-plot approach that leads to 30 min chamber closure time.

Thus, we believe that our approach to describe the observed spatial and temporal variation and covariation in d13C and Fch4, together with interpretation based on our conceptual framework does provide novel insights to methane emissions from mire systems.

**Specific comments:**

*Additionally, there are a few issues that are reducing the clarity of the authors' message. One is the use of the phrase "trophic status", which is used in the manuscript to indicate both seasonal build-up of plant-derived carbon as well as the metabolic pathway of methanogenic archaea. Neither of these is likely the default interpretation that readers will be using when they first encounter the phrase. Distinct phrases should be used (and explained on first use) for these two phenomena and the link between them should be made explicit.*

We agree that the terminology used in the original submission was somewhat confusing and even misleading in places. We used the term "trophic status" in the same as in Hornibrook and Bowes, 2007 and Hornibrook 2009 (references in the manuscript). We imply that the trophic status ( = quality and quantity of available substrates) has an effect on the metabolic pathway. Furthermore, we discussed in places exclusively on shifts between acetoclastic hydrogenotrophic methanogenesis but did not mention the changes in energetics of the hydrogenotrophic methanogenesis, which can facilitate similar relations. Thus, we have revised the terminology and discussion, for them to be more exact. in most places "trophic status" has been replaced with "substrate availability".

*Another issue is the description of the Keeling plot method, which currently leaves the reader to put together that the mixing ratio is based on the up-scaled land cover values, unless my interpretation is widely off-base (see L168 & 219). Please clarify this.*

The Keeling-plot method itself does not need up-scaled land cover values. In the Keeling plot method, the best fit line between d13C and inverse of the CH4 mixing ratio (X) is extrapolated to 1/X=0, as explained in the manuscript (lines 190-198). This is done separately for each chamber closure in chamber approach, and for each night in the nocturnal boundary-layer approach. We have added a reference Rinne et al. (2021) into chapter Methods, giving description of nocturnal boundary-layer Keeling plot approach.

The land cover values are used in upscaling the chamber d13C values (Eq. 2) to represent the whole mire, and for comparison with NBL-A method.

*Finally, the formatting for different taxonomic levels is none-standard and inconsistent throughout the manuscript.*

We have edited the formatting the taxonomic levels following the guide for authors of Biogeosciences.

*Technical corrections:*

*66 Replace "mires' with "wetlands", as this statement applies to all wetlands ecosystems, rather than mires specifically*

We have replaced this in the revised manuscript.

*71 This implies the phase-change fractionation leads to biological (or kinetic) fraction, which is not true. It would be more accurate to describe biotic and abiotic fraction processes as just that, two separate chemical phenomena*

We have reformulated this sentence in the revised manuscript.

*74 Introduce the reader to what makes a mire ecosystem distinct from other wetland types here*

The definition of mire ecosystems has been added to the Introduction of the revised manuscript.

*93 Alternative to what?*

We have replaced "alternative" with "different".

*175 Is this plot within the chamber itself or around the chamber?*

We have replaced "plot" with "chamber" in this paragraph to make it less ambiguous.

***189 Define RMSE***

This is defined in the revised manuscript.

***207-214 unit formatting, "spectrometer"***

These are now corrected.

***239 Totally fine to use gDNA as shorthand for "genomic DNA", but it should be defined on first use.***

This is defined in the revised manuscript.

***268 Spatial and temporal fluxes/signatures as well? This is unclear because of the lack of a separate statistical analysis section***

No, this refers to genetic analysis. The fluxes and isotopic signatures are analyzed with MatLab as is now stated in the revised manuscript.

***Figure 8 takes up a lot of space it is, without providing a lot of information. The 10 day periods could be further collapsed into 4 blocks throughout the growing season, perhaps with a trend line. Or a subset of representative panels could be shown and the remainder moved to a supplement.***

We moved this figure to Appendix, and replaced the Figure 8 with examples and development of r2 and p-value.

***374 Is this mire nearby, how different is the climate/vegetation? Some quantitative context for the comparison would be helpful. Also, is an overlap of one methanogenic genus meaningful? It seems likely be mere chance.***

The Abisko-Stordalen mire is in Northern Sweden and thus very different in its climate. We have discussed these differences in lines 376-380. We have added coordinates to Abisko-Stordalen in the revised manuscript to make the geographic difference more obvious. Our statement is a bit misleading, as we observed the same genera of hydrogenotrophic methanogens and the same genus to be dominant. This is now stated in the revised manuscript. We found this similarity in spite of geographic and climatic difference interesting to warrant mentioning.

The thank for the positive and constructive comments. We address the comments (in bolded italics) in detail below.

***General comments***

***The study by Rinne et al. investigates CH4 emission rates and d13C-CH4 values, and the community structure of methanogenic and methanotrophic communities in a poor fen in southwest Sweden. It is one the most detailed investigation to date pairing high temporal resolution upscaled d13C-CH4 values with integrated d13C values of CH4 flux sampled from nocturnal boundary-layer accumulation. The key findings locally are that: (i) the observed spatial and temporal differences in d13C values of CH4 emissions vary systematically in response to environmental conditions, (ii) the spatial range of values (~15 permil) is larger than temporal variations and appears to be governed by differences in substrate and moisture levels within the peatland that can be identified by vegetation assemblages that can be delineated via remote sensing, and (iii) metagenomic analysis indicates that methanogenic communities within the peatland are diverse and capable of adapting to changes in substrate supply and environmental conditions. I support publication of this work with minor revision.***

***I recommend that the authors explore further in the Discussion section the implications of their measured d13C values for isotope-weighted global CH4 budgets. The measured d13C values (~ -81 to -79 permil) of CH4 emissions from the site are significantly more negative than d13C values typically attributed to global and northern wetlands (e.g., -58‰; Mikaloff-Fletcher et al., 2004a,b; -58‰, Bousquet et al., 2006; -59‰, Monteil et al., 2011;). Similar to Fisher et al. (2017), this study presents further compelling evidence for a need to adjust d13C values attributed to CH4 emissions from northern peatlands.***

We have included a discussion of the mire-scale d13C values in the perspective of d13C values observed in other mires (Fisher et al., 2017 and Menoud et al., 2022), with consequences to the atmospheric inversion modelling of methane sources, in the revised version of the manuscript (Discussion, Conclusions, Abstract).

***Specific comments***

***Manuscript title: '...variation of d13C values of methane...'***

We have changed the title of the manuscript accordingly.

***Line 13 – '...offer clues...'?***

This typo has been corrected.

***Line 76-77 and elsewhere. Replacing terms such as 'isotopically lighter CH4' with more specific language would eliminate the need for clarifying statements in parentheses. For example (lines 75-76) could be written as ' ... hydrogenotrophic methanogenesis typically produced CH4 that is 13C-depleted relative to CH4 generated from acetoclastic methanogenesis.'***

We have edited the text as suggested here. Mostly we have used "lower" and "higher d13C value".

**Line 108: 'reflect differences in CH4 production due to differences in substrate availability for methanogenesis."**

We have changed the sentence accordingly.

**Line 110 and elsewhere: 'methanotrophy prefers 12C, leaving more 13C to the emitted CH4" ='Enzymatic reactions associated with methanotroph metabolism consume 12CH4 preferentially, resulting in 13C-enrichment of residual CH4.'**

We have changed the sentence accordingly.

**Line 113 – awkward sentence; 'less 13C depleted CH4' = '13C-enriched CH4' or 'CH4 having more positive d13C values'.**

We have edited the sentence and used "higher d13C value".

**Line 121 – In this context 'substrate supply' rather than 'trophic status' perhaps would more accurately describe the environmental variable impacting CH4 emission rates.**

We have replaced the "trophic status" with "substrate availability".

Furthermore, our sentence, "…the seasonal cycle of the CH4 emission rate is due to the changes in trophic status, i.e. between acetoclastic-dominated (AM) and hydrogenotrophic-dominated (HM) methanogenesis." implies only changes between acetoclastic and hydrogenotrophic pathways, while also changes in energetics of hydrogenotrophic methanogenesis can work in the same way. Thus, we have also changed this sentence to reflect this reasoning.

**Line 163 – remove capitalization 'polymethyl...'**

This has been done.

**Lines 203-205 – How was the CRDS calibrated in the field for concentration and stable isotope measurements?**

We took parallel samples from chamber closures and run these with IRMS, as explained in the next paragraph. We also occasionally have run standard gas to check the concentration measurement.

**Lines 231-233 – Data from chamber 3 are not mentioned?**

As there is very little data from chamber 3, and its contribution would be low due to small fluxes. We did not include it to the upscaling calculation. We have now mentioned this explicitly in the revised version of the manuscript.

**Line 276 – '...seems to be quite similar...' If this is an important point, perhaps employ a statistical comparison?**

This is actually not an important point for this study, just an interesting observation. Thus, we have removed this sentence from the revised version of the manuscript, as it may confuse a reader.

**Line 304 – 'there were hardly any data'**

This has been corrected it.

References

Menoud, M., van der Veen, C., Lowry, D., Fernandez, J. M., Bakkaloglu, S., France, J. L., Fisher, R. E., Maazallahi, H., Stanisavljević, M., Nęcki, J., Vinkovic, K., Łakomiec, P., Rinne, J., Korbeń, P., Schmidt, M., Defratyka, S., Yver-Kwok, C., Andersen, T., Chen, H., and Röckmann, T.: Global inventory of the stable isotopic composition of methane surface emissions, augmented by new measurements in Europe, Earth Syst. Sci. Data Discuss. [preprint], https://doi.org/10.5194/essd-2022-30, in review, 2022.

*Bousquet, P., Ciais, P., Miller, J.B., Dlugokencky, E.J., Hauglustaine, D.A., Prigent, C., Van der Werf, G.R., Peylin, P., Brunke, E.G., Carouge, C., Langenfelds, R.L., Lathiere, J., Papa, F., Ramonet, M., Schmidt, M., Steele, L.P., Tyler, S.C. and White, J. (2006) Contribution of anthropogenic and natural sources to atmospheric methane variability. Nature 443, 439-443.*

*Fisher, R. E., France, J. L., Lowry, D., Lanoisellé, M., Brownlow, R., Pyle, J. A., et al. (2017). Measurement of the 13C isotopic signature of methane emissions from northern European wetlands. Global Biogeochemical Cycles, 31, 605–623.*

*Mikaloff Fletcher, S. E., Tans, P. P., Bruhwiler, L. M., Miller, J. B., and Heimann, M. (2004a). CH4 sources estimated from atmospheric observations of CH4 and its 13C/12C isotopic ratios: 1. Inverse modeling of source processes. Global Biogeochem. Cy. 18:GB4004, doi:10.1029/2004GB002223.*

*Mikaloff Fletcher, S. E., Tans, P. P., Bruhwiler, L. M., Miller, J. B., and Heimann, M. (2004b). CH4 sources estimated from atmospheric observations of CH4 and its 13C/12C isotopic ratios: 2. Inverse modeling of CH4 fluxes from geographical regions. Global Biogeochem. Cy. 18:GB4005, doi:10.1029/2004GB002224.*

*Monteil, G., Houweling, S., Dlugockenky, E. J., Maenhout, G., Vaughn, B. H., White, J. W. C., and Rockmann, T. (2011). Interpreting methane variations in the past two decades using measurements of CH4 mixing ratio and isotopic composition, Atmos. Chem. Phys., 11, 9141–9153.*

---

## Author Response (AR2)

Authors response to the comment raised by referee #1

*Rinne et al have done an excellent job of addressing my concerns on their manuscript, which analyzes temporal and spatial patterns of mire methane flux, its 13C signature, and potential drivers. I recommend that this interesting and data-rich manuscript be accepted for publication with minor revision. My remaining suggestion is to add ranges of published delta values for acetoclastic and hydrogenotrophic methane emissions and refer to these ranges in both the intro and discussion as appropriate. This will strength the current conclusion that variability is mostly due to shifts in methanogenic pathway, as well as give context to the mire-level values, which are some of the lowest reported.*

We thank for the kind comments and very detailed and constructive comments on the earlier version of this manuscript, which helped improve it considerably.

We have added the ranges of delta values for acetoclastic and hydrogenotrophic methane, as suggested.

*Technical edit:*

*L359 "Following double root transformation…", perhaps square root is meant?*

Yes indeed, we have now corrected this.

*General comment:*

*While this reviewer chooses to remain anonymous, she would like to draw brief attention to the assumption about gender identity made in the response to reviewer. Gentle reminder that "the reviewer" and "they" are the safe choices in unknown situations.*

Sorry for this mishap, I try always to be inclusive and neutral in such statements. Coming from a fenno-ugric linguistic background, with no gendered pronouns, I have messed the personal pronouns of indo-european languages more than once. (My own preferred personal pronoun would be the Finnish "hän" which means both he and she).